# Mechanistic analysis of enhancer sequences in the estrogen receptor transcriptional program
Shayan Tabe-Bordbar [1], You Jin Song[2], Bryan J. Lunt[1], Zahra Alavi[3], Kannanganattu V. Prasanth[2] & Saurabh Sinha [4] ✉

Estrogen Receptor α (ERα) is a major lineage determining transcription factor (TF) in mammary gland development. Dysregulation of ERα-mediated transcriptional program results in cancer. Transcriptomic and epigenomic profiling of breast cancer cell lines has revealed large numbers of enhancers involved in this regulatory program, but how these enhancers encode function in their sequence remains poorly understood. A subset of ERα-bound enhancers are transcribed into short bidirectional RNA (enhancer RNA or eRNA), and this property is believed to be a reliable marker of active enhancers. We therefore analyze thousands of ERα-bound enhancers and build quantitative, mechanism-aware models to discriminate eRNAs from non-transcribing enhancers based on their sequence. Our thermodynamics-based models provide insights into the roles of specific TFs in ERα-mediated transcriptional program, many of which are supported by the literature. We use in silico perturbations to predict TF-enhancer regulatory relationships and integrate these findings with experimentally determined enhancer-promoter interactions to construct a gene regulatory network. We also demonstrate that the model can prioritize breast cancer-related sequence variants while providing mechanistic explanations for their function. Finally, we experimentally validate the model-proposed mechanisms underlying three such variants.

Breast cancer is the second most common cancer in women in the United States, with about 12.4% of women diagnosed with breast cancer during their lifetime[1]. The most common breast cancer subtype is characterized by increased activity of Estrogen Receptor α (ERα), a protein that is activated by estrogen and in turn changes the expression of hundreds of genes. Acting as a transcription factor (TF), it binds to thousands of DNA locations and influences the expression levels of nearby genes[2]. This cascade of events is called the "ERα transcriptional program", and aberrations in this program lead to increased cell proliferation and cancer. Several drugs have been developed to treat the ER+ subtype of breast cancer by reversing aberrations in the ERα program or interfering with its cancer-causing effects. However, about 50% of treated patients either do not respond to or develop acquired resistance against these drugs[3]. As such, there is great interest in characterizing the major principles and crucial details of the ERα transcriptional program.

The regulatory information controlling gene expression programs are known to be in part stored in relatively short stretches of DNA (between 1–2 kbp) called enhancers[4]. Enhancers typically harbor several binding sites for one or more transcription factors (TFs). Each binding site is an approximately 10 base long DNA sequence with relatively high binding affinity for its corresponding TF. Active enhancers—marked by specific chromatin marks—regulate expression of genes in their spatial proximity through binding to a specific set of TFs. Notably, the spatial proximity of a gene-enhancer pair depends on the chromatin conformation and is not necessarily captured by their genomic distance. In other words, genes can be regulated by enhancers located virtually anywhere on the genome. ERα is known to mainly regulate expression of genes through binding to distant enhancers, as only about 5% of its binding sites are located within a distance of 5 kb from Transcription Start Site (TSS) of any gene[5].

Over the past decade, biotechnological advancements such as RNA-sequencing (RNA-Seq) and chromatin immunoprecipitation coupled with sequencing (ChIP-Seq) have dramatically expanded our knowledge about the ERα transcriptional program. RNA-Seq assays provide data on expression levels of all genes in different experimental conditions, while

[1]Department of Computer Science, University bof Illinois at Urbana-Champaign, Urbana, IL, USA. [2]Department of Cell and Developmental Biology, University of Illinois at Urbana-Champaign, Urbana, IL, USA. [3]Department of Physics, Loyola Marymount University, Los Angeles, CA, USA. [4]Department of Biomedical Engineering, Georgia Institute of Technology, Atlanta, GA, USA. ✉e-mail: saurabh.sinha@bme.gatech.edu

ChIP-seq experiments reveal all the locations in the genome where ERα is bound in those conditions. Due to its biological importance and potential impact, the ERα transcriptional program has been studied not only through mainstream experimental approaches such as RNA-seq, ChIP-seq, and DNase-Seq[6,7], but also relatively more advanced and expensive experimental techniques such as Chromatin Interaction Analysis by Paired-End Tag Sequencing (ChIA-PET)[8,9], Global run-on sequencing (GRO-Seq)[10–13], and high-resolution chromatin conformation capture (Hi-C)[14], resulting in an unprecedented wealth of diverse data describing this system.

Here, we aimed to comprehensively characterize gene regulatory relationships in this system through modeling of gene expression data, taking advantage of several of the above-mentioned multi-omics data. A common approach to reconstruction of gene regulatory networks (GRN) is to train a quantitative model that correctly predicts gene expression levels, and then interpret the parameters of the model to identify regulatory relationships. There exist two major variations of the gene expression prediction problem. In the "sequence-to-expression" formulation, DNA sequences (enhancers) associated with a gene should bear the "footprints" of the TFs that ostensibly regulate the gene, since a TF needs to physically bind to enhancers in order to regulate the gene (Supplementary Fig. 1). In the alternative, less constrained "expression-to-expression" formulation, one attempts to relate a gene's expression to the expression of its potential regulators, without taking DNA sequence information into account. In the current study, we considered the former formulation of the expression prediction problem in order to study the ERα transcriptional program, reconstruct the underlying GRN, and decipher the sequence-level encoding of the network.

Sequence-to-expression modeling in mammalian systems poses two key challenges: enhancer-gene assignment, and enhancer-readout prediction. In enhancer-gene assignment, one must identify the genomic location of enhancers regulating a gene's expression and their relative contributions in a particular condition. On the other hand, enhancer-readout prediction amounts to quantifying the expression driven by a particular enhancer in a given cellular condition, using its sequence and the abundance of its regulators in that condition. Experimental approaches such as high-resolution Hi-C and ChIA-PET detect enhancer-promoter interaction and can help address the enhancer-gene assignment problem[15]. In contrast, reporter assays remain the primary technique to quantify enhancer-driven transcriptional activity. Genome-integrated reporter assays are generally low-throughput and laborious, hence are a less practical source of ground truth data for enhancer-readout modeling exercises. However, it has been recently discovered that short bidirectional RNA is transcribed from a subset of regions with active enhancer marks such as the histone modifications H3K4me1 and H3K27Ac[16]. Observation of these somewhat mysterious transcripts—called enhancer RNA (eRNA)—has been associated with active enhancers, increased expression of nearby genes and enhancer-promoter looping[16–18]. Since eRNA transcription is mechanistically similar to mRNA transcription, correlates with expression of nearby genes, and can be detected in high-throughput experiments such as GRO-Seq and RNA-Seq, it can serve as the measure of enhancer-driven transcriptional activity. The exact role of eRNA is currently an active area of research, in particular among breast cancer researchers, since eRNA presence and transcription seem to be specially related to enhancer-promoter loop formation, and ERα is known to mainly act on target genes through loop formation with distant enhancers[10,19,20].

In a pioneering study, Li et al.[10] observed that when MCF7 cells (ERα+ breast cancer cell line) are treated with estrogen, thus activating ERα, of all the ERα-bound genomic regions with the active enhancer mark H3K27Ac, only a subset are transcribed (eRNAs). These enhancers generally show stronger H3K27Ac signal[20], are enriched near upregulated genes[10], and mark functional TF binding, singling them out for their potential regulatory function. However, it is not clear why this specific subset of enhancers is transcribed, and this is the central problem addressed in our study. We sought to identify the set of sequence features that distinguish ERα-bound enhancer regions that are transcribed from those that are not, thereby

probing the mechanisms underlying ERα- driven regulation. The unique nature of eRNAs as seats of regulatory information as well as products of regulation makes them ideally suited for sequence-to-expression modeling. By limiting our analysis to ERα-bound enhancer regions, we expected to identify cis-regulatory features beyond the most obvious ones, i.e., ERα-DNA binding and the H3K27ac mark. We first used a Random Forest (RF) classifier to extract sequence features (TF motifs) distinguishing the above-mentioned eRNAs and applied the identified features in a thermodynamics-based model to glean mechanistic insights into ERα transcriptional program and its cis-regulatory encoding.

Thermodynamics-based frameworks have been previously applied to predict the expression driven by enhancers in model organisms where enhancer-readout data are available through reporter assays[21–23]. In this study, we used a thermodynamics-based model called GEMSTAT[24] to predict transcription of enhancers (eRNAs) during estrogen-mediated transcriptional response in breast cancer cells. As a result of this exercise, we learned mechanistically interpretable parameters for each TF quantifying its effect on any enhancer. We also modeled the interactions between these TFs and explored how such interactions affect the models. Identified roles for the majority of the TFs are supported by literature evidence. We next used the trained models to predict the effect of trans and cis perturbations. In particular, we performed in silico knock-down experiments for all considered TFs, thus predicting targets of individual TFs and constructing a TF-enhancer regulatory network for this system. Enhancers predicted to be regulated by the TF AP2-γ were supported by available GRO-seq data in matching conditions. Additionally, in order to investigate cis perturbations, we ranked breast tissue/cancer-related sequence variants by their effect predicted using the learned model, and demonstrated the higher precision of such prioritizations as compared to baselines that rely solely on epigenomic information. Our models provide TF-centered mechanistic explanations for alterations of enhancer activity due to genetic variations. Finally, we chose three potentially functional variants, proposed a mechanistic explanation underlying their effect, and experimentally tested our hypothesis.

## Results

### Sequence features of transcribed ERα binding sites

This study follows the workflow described in Fig. 1. We studied genomic regions (Fig. 2a) that are bound by the ERα transcription factor and at the same time exhibit the active enhancer mark H3K27Ac upon treatment of the MCF7 cell line with estrogen, as identified via ChIP-seq experiments by Li et al.[10]. We sought to determine the sequence features that distinguish the subset of these regions that are transcribed (eRNAs; 489 regions shown in light green in Fig. 2a) from those that are not (3245 regions shown in orange in Fig. 2a); eRNAs are known based on GRO-Seq experiments from the same study. We expected these features to provide highly specific clues into the cis-regulatory logic controlling direct regulatory targets of ERα, to be used subsequently for mechanistic modeling. We thus faced a classification problem involving 3734 genomic segments that can potentially act as active enhancers while directly interacting with the active form of ERα. Focusing specifically on enhancers affected by estrogen, we excluded 40 out of 489 eRNA-overlapping examples shown in Fig. 2a because those regions overlapped with GRO-Seq peaks both before and after estrogen treatment. Additionally, keeping the noisy nature of GRO-Seq data in mind, to avoid false negatives in our classification analysis we excluded 1576 out of 3245 non-eRNA-overlapping examples shown in Fig. 2a because of overlap with GRO-Seq peaks in one of the two replicate experiments. Hence, we designated 449 of these as "positive" examples (transcribed only upon estrogen treatment), 1669 as "negative" examples (not transcribed upon treatment) and ignored the remaining. Coordinates (hg38 genome version) of positive and negative class enhancers is described in Supplementary Data 1.

We next used a Random Forest (RF) classifier to discriminate the positive and negative sets of segments based on their sequence features. These features included computationally predicted binding sites of 30 potential regulators (TFs) of the ERα transcriptional program, as well as

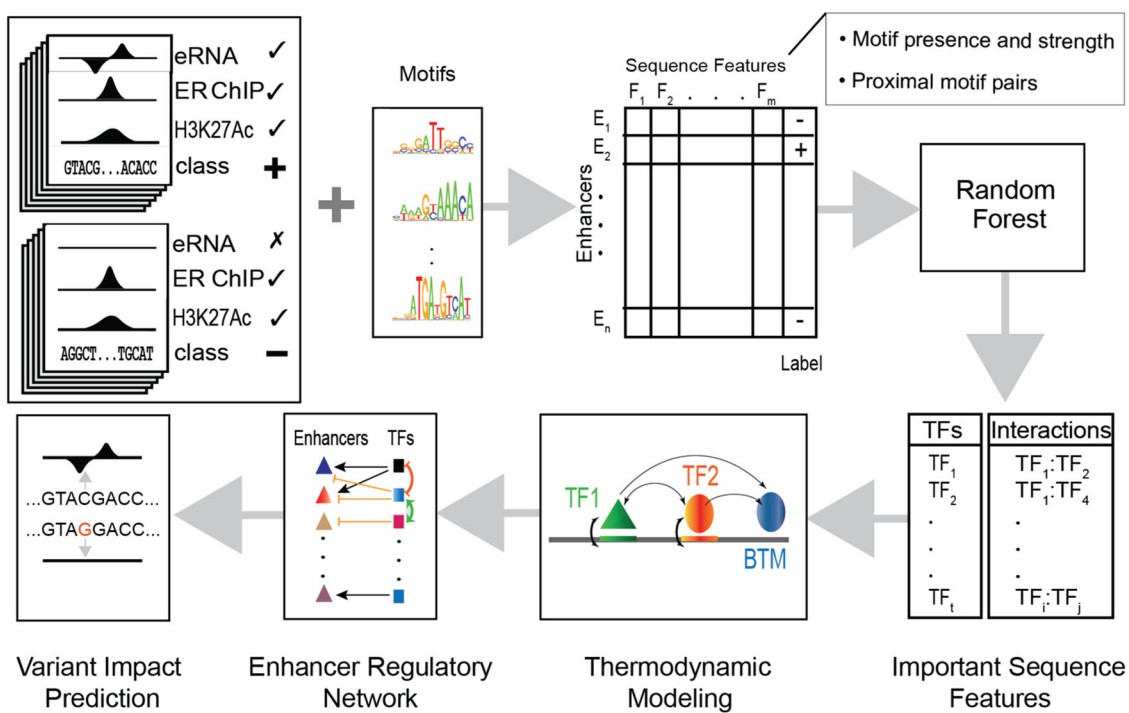

**Fig. 1 | Schematic overview.** Genomic regions marked by ERα ChIP peaks and H3K27Ac ChIP peaks were divided into positive/negative sets based on the presence/absence of eRNA signal from GRO-seq data. Enhancer sequences from both classes were scanned using motifs of relevant TFs to quantify binding affinity. Additionally, the number of adjacent sites (within 50 bp) for each pair of TFs was counted. Motif- and motif pair-based features thus defined were used in a Random Forest (RF) classifier and important features were extracted. Thus, identified TFs and their pairwise interactions were used to train a thermodynamics-based model. The trained model led to an Enhancer Regulatory Network (through in silico knock-downs of TFs) as well as predictions of regulatory variants (through expression prediction on enhancers with alternative alleles).

their co-occurrence information. A total of 594 numeric features were used to describe each segment, of which 33 represented individual TF affinities and 561 represented adjacent binding sites of pairs of TFs (see "Methods"). Several of the defined features were noted to have a statistically significant difference (t-test p value < 0.05) between positive and negative classes (Fig. 2b, c). Overlapping histograms of normalized TF affinity scores for all considered TFs is shown in Supplementary Fig. 2. After partitioning the segments into training and test sets in the ratio of 4:1 (approximately), we trained the RF classifier on the training set and evaluated its performance on the test set, observing an Area Under Receiver Operating Characteristic curve (AUROC) of 0.70 and an Area Under Precision-Recall Curve (AUPRC) of 0.39 (Fig. 2d, e). These performance metrics are substantially greater than random baseline expectations of 0.5 (AUROC) and 0.21 (AUPRC) respectively, indicating that the model has predictive power. We assessed the importance of each feature using standard methods (see "Methods") and the 16 most important features (Supplementary Table 1) which included motif score for 14 TFs and pairwise adjacency scores for two pairs of motifs, were selected to be included in the next phase of the study.

**Thermodynamics-based modeling of enhancers**
The RF classifier learns important sequence features that characterize functional enhancers (ERα-bound eRNAs) mediating the ERα transcriptional response. However, it does not directly address mechanistic questions regarding the system: an ensemble of decision trees voting to predict the class label (transcribed or not) is not easily mapped to a mechanistic explanation of how expression is encoded in enhancer sequences. For instance, the trained RF model reported above utilizes votes from 1000 different decision trees, each decision tree comprising on average 167 nodes or "decisions", making it nearly impossible to interpret the model in biologically meaningful ways. Similarly, the model does not offer us insights into whether a TF pair with co-occurring sites, if found to be a predictive feature, interact synergistically or antagonistically. In contrast, biophysics-inspired models that predict expression readout of an enhancer based on molecular interactions involving TFs and DNA binding sites are well-suited for mechanistic understanding of enhancer function and have been used successfully in modeling of combinatorial regulation in developmental enhancers[21,24–26]. We therefore analyzed the above set of 2118 genomic segments using the thermodynamics-based model GEMSTAT[24] to predict expression from sequence, utilizing the RF-identified sequence features that represent TFs and TF pairs most important to the system.

GEMSTAT first detects putative TF binding sites present in an enhancer sequence using the TFs' binding motifs, then interprets the collection of putative sites using a biophysical framework to predict the activity of the enhancer. The model uses only two tunable parameters per TF, with clear biophysical interpretations, and a free parameter for each TF pair hypothesized to interact when DNA-bound. We customized the GEMSTAT model implementation for this work: the quantitative expression level assigned by GEMSTAT to each enhancer is mapped to the probability of that enhancer being a positive or negative example (eRNA or not) and model parameters are then trained so as to maximize the likelihood of data (see "Methods").

We chose 14 most highly ranked TFs based on Random Forest importance score (Supplementary Table 1), together with three other TFs known to be involved in the ER transcriptional program (i.e., GATA3, GR, and RARα) to form the set of TFs used for GEMSTAT modeling. Additionally, based on the literature and the trained random forest model, we allowed nine TF-TF interaction terms to be included in the GEMSTAT models. These interaction terms consist of interactions of ERα with its potential co-factors FOXA1, GATA3, PBX1, PGR, RARα, AP2-γ, together with a self-interaction pair for JUN-1 allowing for its self-cooperativity, as well as two most highly ranked TF-TF pairs in terms of importance score by the random forest models (i.e., ERα-YBX1, and RUNX1-YBX1). Using the 17 noted TFs and the 9 mentioned TF-TF interactions (culled from a larger collection of 30 TFs and 561 TF pairs during RF classification), we trained an

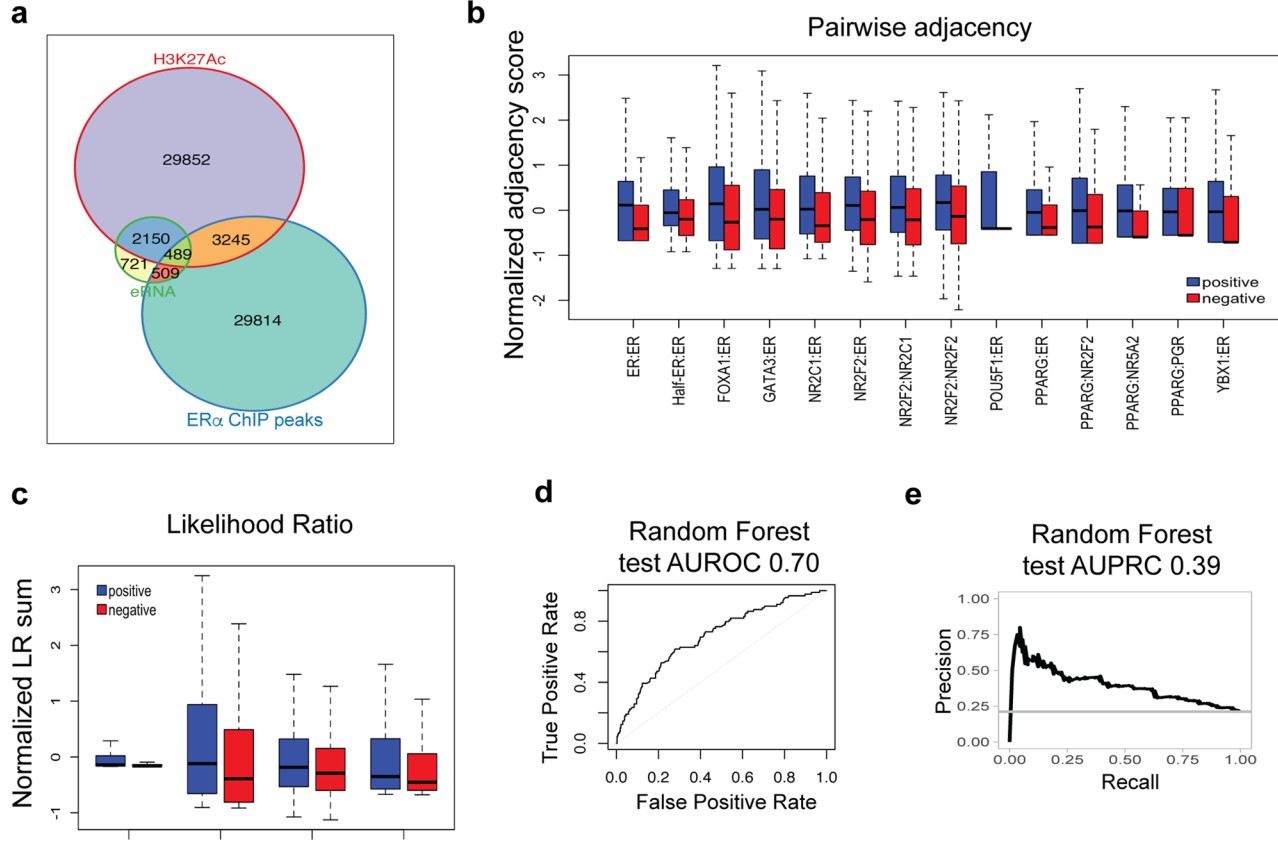

**Fig. 2 | RF classification. a** Venn diagram of datasets used in this study. Regions colored orange (3245) and green (489) make up the universe set of our classification examples. The green region contains positive examples (449 out of 489, see "Methods"), and orange region contains negative examples (1669 out of 3245, see "Methods"). **b** Boxplot illustrates pairwise adjacency scores for a selection of TF pairs with statistically significant difference between positive and negative sets (*p* value < 0.05). Blue and red boxes represent enhancers in the positive and negative sets, respectively. *Y*-axis represents the *z*-score normalized pairwise adjacency score, which reflects the frequency of co-occurrence of motifs of the TF pair within 50 bp of each other. **c** Boxplot shows TF affinity scores for four TFs with significantly different affinity scores between positive (blue) and negative (red) sets of enhancers (*p* value < 0.05). *Y*-axis represents *z*-score normalized sum of Likelihood Ratios under the PWM model versus the background model. **d** Receiver operating characteristic curve indicating the performance of Random Forest classifier on the test set. **e** Precision-Recall curve shows performance of Random Forest classifier on the test set. Note that for all the boxplots in this manuscript center line indicates median; box limits are upper and lower quartiles; whiskers show 1.5x interquartile range, and outliers are removed to aid visualization.

ensemble of 4624 GEMSTAT models. Each model in the ensemble is an assignment of values to the free parameters of GEMSTAT, such that the model predictions match training data. (We have previously shown the advantages of ensemble modeling as a means to address potential issues of identifiability[27]). Models were trained on 200 positive examples and 600 negative genomic segments, then filtered using a validation set containing 90 positive and 334 negative examples. Figure 3a, b illustrates training and validation performance for all models in terms of AUROC and AUPRC. We chose the union of top 150 models based on either AUPRC or AUROC criteria to work with for the rest of this study. (A total of 244 models were thus selected, shown as red points in Fig. 3a, b). Figure 3c shows the performance of the selected models (henceforth called the "final ensemble") on an unseen test set containing 89 positive and 333 negative examples (same test set as that used in Random Forest evaluations). As an example, the performance of the model with the highest test AUPRC is depicted in Fig. 3d, e (AUROC = 0.66, AUPRC = 0.40). This and other models in the final ensemble have test performance comparable to the RF classifier (above), while utilizing far fewer parameters and a biologically interpretable model structure and were thus the basis of our further investigations.

We inspected models in the final ensemble by visualization of their learned parameters, as shown in Fig. 3f and Supplementary Fig. 3. These include a "binding parameter" (reflecting strength of TF binding to its optimal site) and an "activation parameter" (reflecting strength of activation or repression by a single enhancer-bound TF molecule) for each TF, as well

as a "cooperativity parameter" (reflecting the strength and type of interaction—synergistic or antagonistic) for each TF pair assumed to interact when bound to proximal sites. Overall, the 244 models in the ensemble are highly similar to each other, suggesting that they have consistent interpretations. For instance, all models assign a strong activation role to ERα, PAX2 and RARα and most models assign a repressive role to Glucocorticoid Receptor (GR). Similarly, almost all models learn a strong cooperative interaction between ERα and RARα. On the other hand, some differences exist among the ensemble models allowing us to cluster them into three groups. Model differences include ones pertaining to the nature of interaction (synergistic or antagonistic) for the TF pairs (RUNX1, YBX1) and (ERα,YBX1), and the role (activating or repressive) assigned to PGR. Models in groups 1 and 3 (obtained from clustering) learn YBX1 as a weak repressor with strong binding potential that interacts negatively with ERα and positively with RUNX1. On the other hand, group 2 models learn YBX1 to be a strong repressor with relatively weak binding potential, interacting negatively with RUNX1 and positively with ERα. In light of previous studies[28,29], this interplay between ERα, YBX1, and RUNX1 is an interesting subject for future follow-up work. Other than these cases, parameters were qualitatively consistent across the ensemble.

**In silico perturbations reveal TF roles and target enhancers**
Going beyond the global examination of TF roles and pairwise interactions, we next sought to characterize the regulatory mechanisms represented by

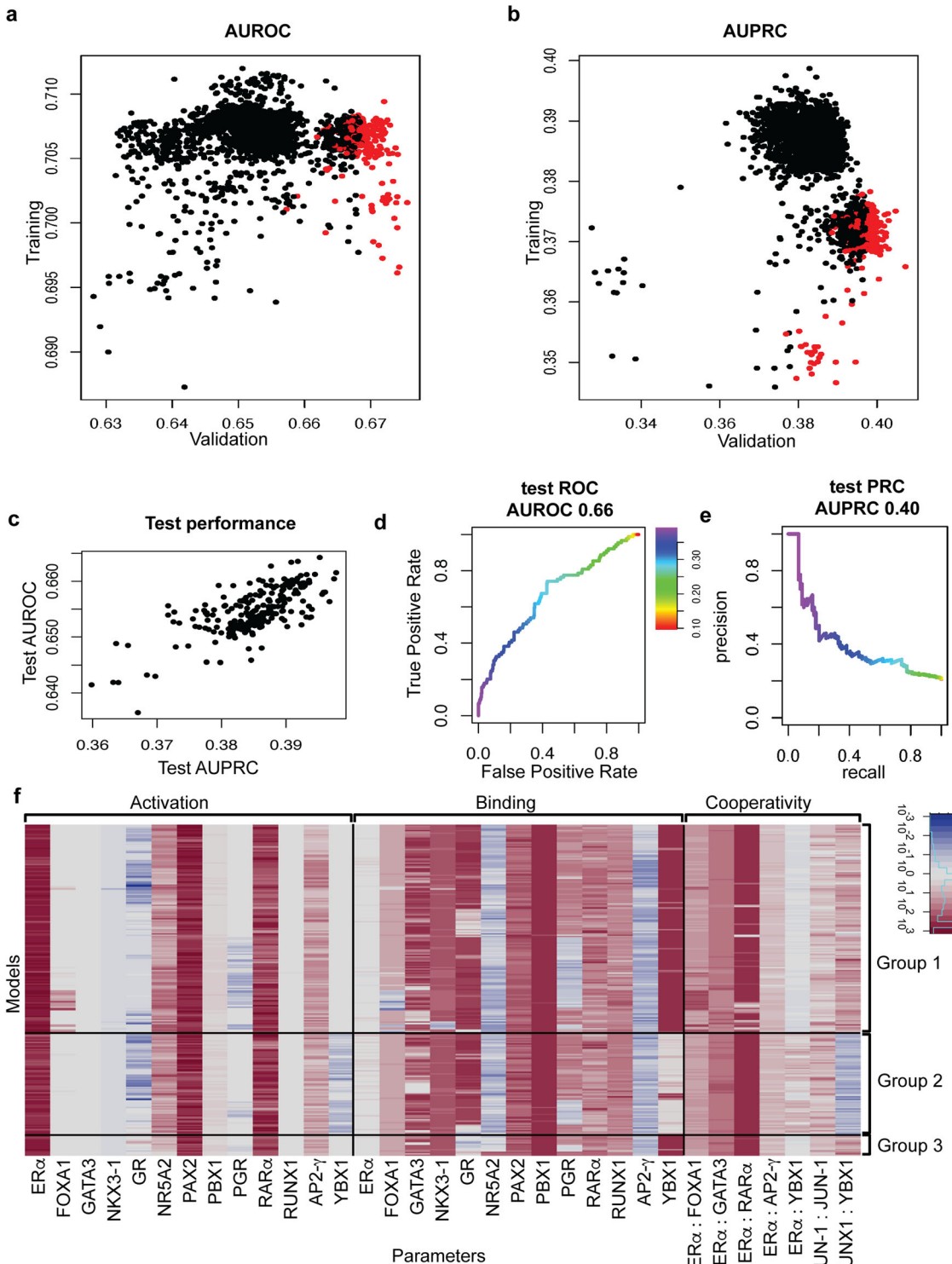

**Fig. 3 | GEMSTAT performance measures and parameters. a** Scatter plot shows the training and validation performance (area under ROC curve, or AUROC) of 4624 GEMSTAT models in constructed ensemble. Each point represents a model, *Y*-axis shows its training AUROC, and *X*-axis shows validation AUROC. Points in red indicate models that are in the top 150 of the ensemble by either the validation AUROC or the validation AUPRC; these models were selected for testing on unseen data (**c**). **b** Same as (**a**), except the performance metric shown is area under PR curve (AUPRC), instead of AUROC; red points have the same meaning as in (**a**). **c** Scatter plot represents the test performance (AUROC and AUPRC) of models colored red in (**a**, **b**). **d**, **e** ROC and PR curves indicating test performance of the best model in terms of AUPRC on the test set. Color bar indicates raw prediction threshold

values used to generate the ROC and PRC curves. **f** Heatmap representation GEMSTAT ensemble model parameters. Rows represent 244 GEMSTAT models forming the ensemble, and falling into three broad clusters. Columns correspond to model parameter values that belong to "Activation", "Binding", or "Cooperativity" categories. Larger values of Binding parameter indicate greater binding potential for a TF. Activation parameter greater/less than 1, represents activatory/repressive role for a TF, respectively and its absolute log10 transformed value represents the regulatory strength of the TF. Cooperativity parameter of greater/less than 1 represents cooperative/antagonistic interaction, respectively, and its absolute log10 transformed value represents the strength of interaction.

**Fig. 4 | In silico perturbations reveal TF target enhancers.** Results of in silico knock-down experiments using a representative GEMSTAT model for ERα (**a**), NKX3-1 (**b**), and ERα-FOXA1 interaction (**c**). **d**, **e** Bars depict the extent to which in silico perturbations of TFs or TF-TF interactions affect enhancers, shown separately for positive (eRNA expressing) and negative (silent) enhancer classes. Bar height represents the average over 244 top ranking models (shown as individual points), and error bars represent the standard deviation. The extent of perturbation effect is shown as (**d**) the fraction of enhancers whose expression changes by at least 5 percentile points in each perturbation experiment, and (**e**) the percentile change in expression after perturbation, averaged over enhancers that were affected by at least 5 percentile points. Pairs with statistically significant difference between positive and negative group enhancers (adjusted Wilcoxon *p* value < 1e−6) are marked with asterisks (*).

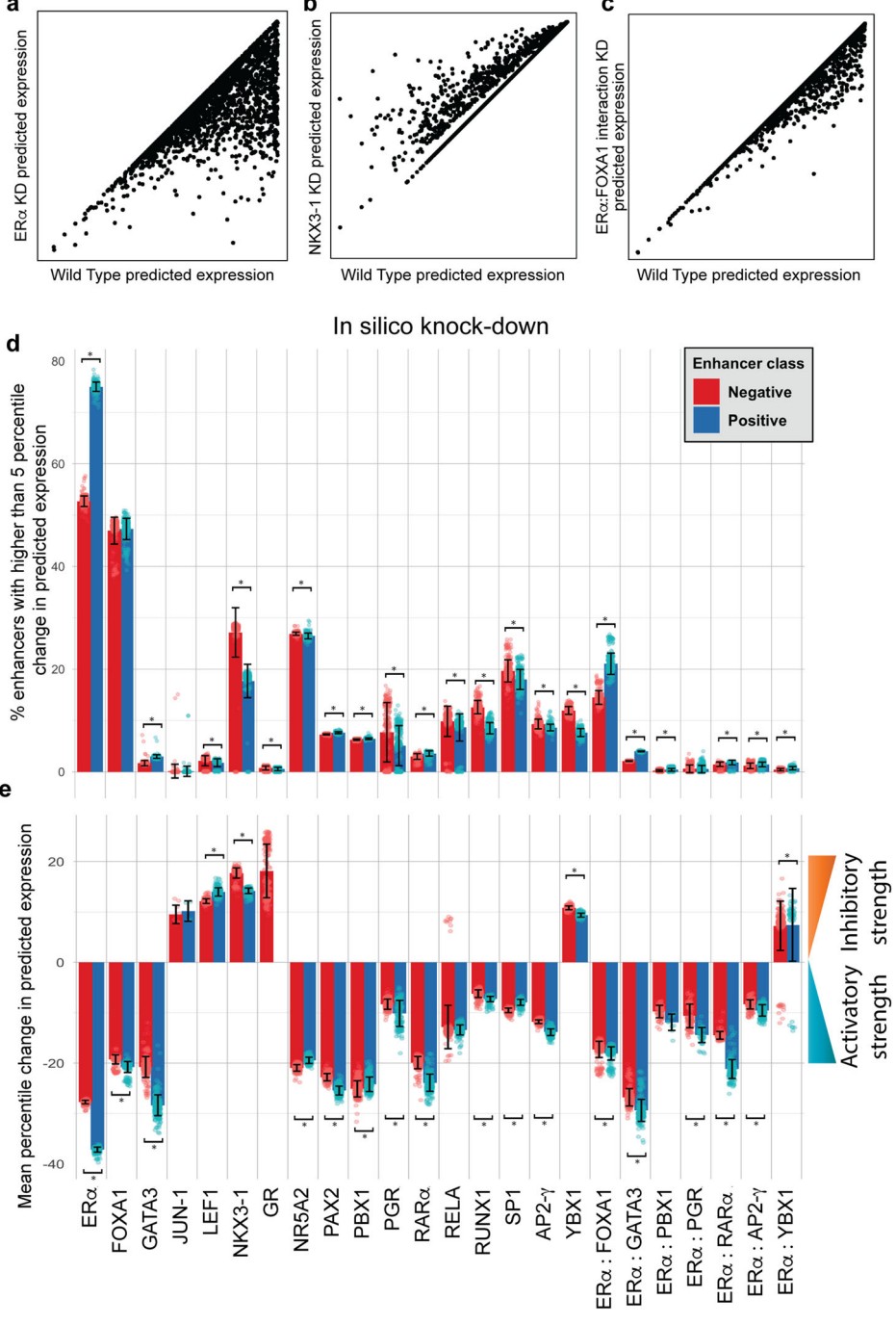

the final ensemble at the level of individual enhancers. In particular, we quantified the predicted effect of a TF on enhancers by performing "in silico knock-down" experiments, where we used trained GEMSTAT models to predict the expression driven by each enhancer in the absence of the TF. We similarly examined the influence of a TF-TF interaction by setting the corresponding parameter to a null value. Note that the models make a numeric prediction for the activity of both positive and negative class enhancers. In case of a perfectly accurate model, the predicted values for all negative class enhancers are less than that of all positive class enhancers. The predicted activity can be compared before and after removal of a certain TF (i.e., in silico knock-down) to quantify the TF's predicted effect on enhancer activity as learned by the models.

Figure 4a–c illustrates the effect of three such in silico perturbations on every enhancer as predicted by a sample model in the ensemble. Note that

the type of influence (activation/repression) of each TF or TF-pair, as per the model, is immediately revealed by this analysis. We performed such in silico perturbations for each model in the final ensemble and examined average expression change predicted by the ensemble for each perturbation. Supplementary Fig. 4 represents a heatmap of the predicted effect of removing each TF on considered enhancers and Supplementary Data 2 contains that information as a spreadsheet.

Figure 4d shows the fraction of enhancers whose predicted expression changes by at least 5 percentile points following each perturbation, compared to the wild-type (WT) prediction. (Recall that GEMSTAT directly predicts expression levels rather than class labels.) Notably, most perturbations affect a small fraction (<10%) of enhancers and as a result the mean change, over all enhancers, is generally small (Supplementary Fig. 5). Hence, we additionally examined the effect of perturbations averaged over only

those enhancers that are affected by the perturbation (≥5 percentile point change, as above) (Fig. 4e). Figure 4d, e clearly illustrates the differential effect of several in silico perturbations on transcribing (eRNA) and silent classes of enhancers. For example, ERα is predicted to affect a larger fraction of eRNA transcribing enhancers ("positive" class shown in blue, Fig. 4d) and the effect size on this class of enhancers is also more pronounced, as compared to silent enhancers ("negative" class shown in red, Fig. 4e). In contrast, both NKX3-1 and YBX1 predicted to influence a larger fraction of silent enhancers than transcribed enhancers and do so more strongly. Furthermore, these two TFs are learnt to have inhibitory effect on their targets (Fig. 4e), consistent with silent enhancers being their main targets.

We next examined the regulatory influence of several TFs and TF pairs, as inferred above, in light of the rich literature on the ERα transcriptional program (Table 1). First and foremost, as expected, ERα is learned as the most common regulator, activating enhancers of positive as well as negative classes, with a larger fraction of positive class enhancers under its influence. (Recall that both classes comprise segments with ERα ChIP peaks and H3K27Ac marks, with the only difference between classes being that the positive class is transcribed as eRNAs.) FOXA1 is perhaps the most well-known co-regulator of ERα[5,30]. As shown in Fig. 4e and consistent with previous findings, our models predict that FOXA1 activates a large fraction (nearly 50%) of examined enhancers, in both classes. It is well-known that FOXA1, as a pioneer factor, facilitates ERα binding through their interaction. This is in agreement with our model-based finding of a strong cooperative interaction between ERα and FOXA1. (This interaction has the most frequent predicted effect among all TF pairs.) Additionally, as illustrated in Fig. 4e, NKX3-1 is identified as a repressor consistently affecting about 30% of negative and about 20% of positive enhancers, potentially counteracting the activating effect of ERα. The TF LEF1 is learned as a repressor with strength similar to NKX3-1 (Fig. 4e) but only influencing a handful of enhancers (Fig. 4d). The learned inhibitory roles of NKX3-1 and LEF1 have been previously suggested and supported by experimental results[31]. Furthermore, the final ensemble consistently identified NR5A2 to be a transcriptional activator for about 30% of the examined enhancers, in agreement with previous reports[32]. YBX1 was learned as an inhibitory factor, mostly affecting the negative class enhancers. Also, YBX1-ERα interaction, despite affecting a small fraction of examined enhancers, consistently represses those enhancers. This physical interaction and its repressive role have also been previously described in literature[33]. As a final example, AP2-γ, a pioneer factor known to have an activating role in the ERα transcriptional program[34], was correctly assigned an activating role by the model. A more complete listing of inferred roles along with supporting literature evidence is provided in Table 1 and Supplementary Table 2. The fact that the majority of the learned roles are in line with previous studies provides external evidence supporting the models.

In some cases, the literature evidence about a TF's role is ambiguous. For instance, opposing effects of RARα and ERα agonists[35], evidence of cooperativity between these two TFs[36], their similar DNA binding preferences, and association of RARα with drug resistance in breast cancer cells[37] make RARα's role in the ERα transcriptional program particularly controversial. Our models learned a strong activation role for this TF and its interaction with ERα, although affecting only a small fraction of enhancers. Progesterone Receptor (PGR) and NFκb (RELA) are two other TFs with contradictory literature evidence regarding their role in the ERα-mediated transcriptional program[13,38–42]; these were found as activators by the majority of ensemble models, affecting 6.4% and 9.3% of the enhancers, respectively. Glucocorticoid Receptor (GR) is another nuclear receptor with unclear role in the studied transcriptional program[43]. This TF was learned as a transcriptional repressor by our models, although it affects only a few enhancers.

The above analysis allows us to derive an "enhancer regulatory network" that connects TFs to enhancers. For instance, AP2-γ knock-down is predicted by the models to significantly reduce the expression of about 9% of tested enhancers (Table 1), thus defining a putative regulon of the TF during estrogen response. We used GRO-seq data[44] in the same cellular conditions (estrogen-treated MCF7 cell lines) obtained in AP2-γ knock-down and

**Table 1 | Literature evidence supporting the roles of TFs and TF pairs, as predicted by GEMSTAT models**

| TF | GEMSTAT inferred role | % enhancer affected | Avg. % change | Literature-suggested role for TF | Reference PMID |
|---|---|---|---|---|---|
| ERα | Strong activator | 63.8 | −32.5 | Transcriptional activator | 26684552 |
| FOXA1 | Weak activator | 47.1 | −20.0 | Pioneer factor. Alters DNA accessibility profile and is necessary for Estrogen-induced ERα binding. | 18358809,16009131, 21151129 |
| NKX3-1 | Weak inhibitor | 22.4 | 16.0 | Represses ERα activity by competitively binding to chromatin/ recruiting HDAC1 | 18794125 |
| NR5A2 | Strong activator | 26.7 | −20.2 | Activates ERα-mediated transcription at least partly through co-binding to Estrogen Response Elements. | 22359603 |
| PBX1 | Weak activator | 6.4 | −24.7 | Pioneer factor | 22125492 |
| PGR | Strong/Weak inhibitor | 6.4 | −9.2 | Unclear/controversial | 26153859,28729413, 27885264 |
| RELA (NFκb) | Ambiguous (weak activator/strong inhibitor) | 9.3 | −13.1 | Transcriptional activation; controversial since ERα is known to repress NFκb transcriptional activity | 25752574,20705611, 19920189,18703630, 22963717 |
| RUNX1 | Weak activator/Strong binding | 10.55 | −6.75 | Mediates indirect binding (tethered) of ER to DNA | 20547749 |
| SP1 | Weak activator | 18.8 | −8.7 | Transcriptional activation in part due to mediation of indirect ERα-DNA binding | 9328340,11250935, 11345900 |
| AP2-γ | Strong Activator | 9.0 | −12.8 | Pioneer factor | 21572391 |
| YBX1 | Weak/Strong inhibitor | 9.8 | 10.1 | Transcriptional repression through direct interaction with ERα | 29180470 |
| ERα:FOXA1 | Cooperative | 17.8 | −17.7 | Cooperative activity essential for ERα activity | 18358809,16009131, 21151129 |

"% enhancer affected" indicates the percent of enhancers affected by ≥5 percentile (ensemble average). "Avg. % change" represents the predicted percentile change in activity after knock-down, averaged over enhancers that were affected more than 5 percentile points. Note that only factors that affect more than 5% of the tested enhancers (i.e., % enhancer affected ≥5) are included in this table. Same information for all TFs and interactions is shown in Supplementary Table 2.

**a**

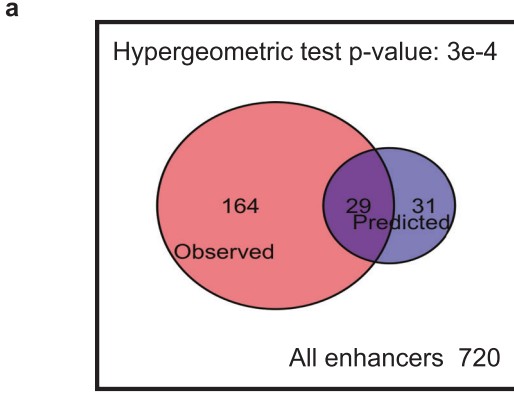

**b**

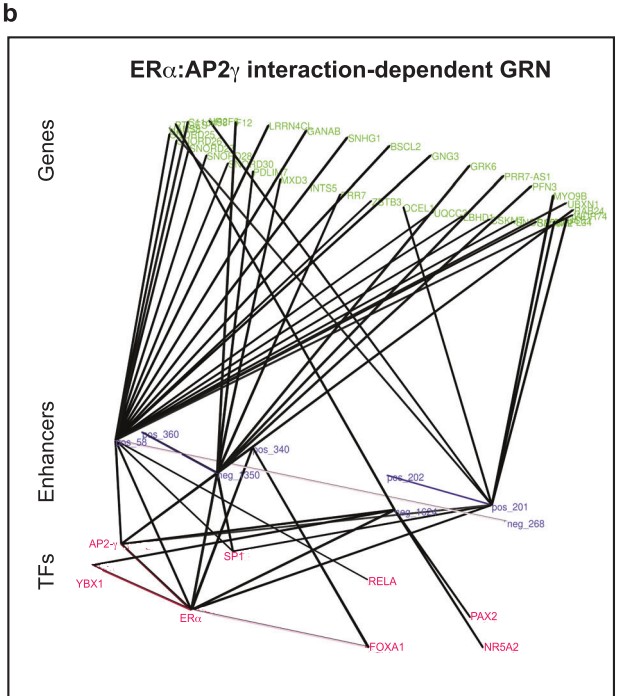

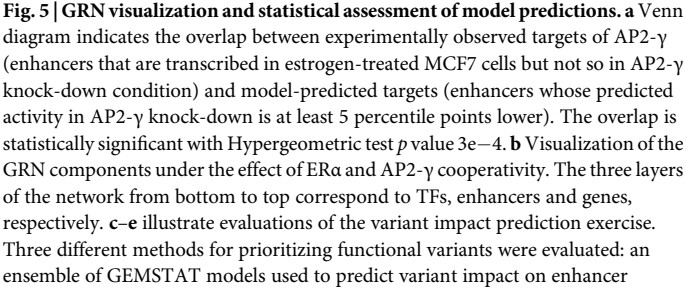

**c**

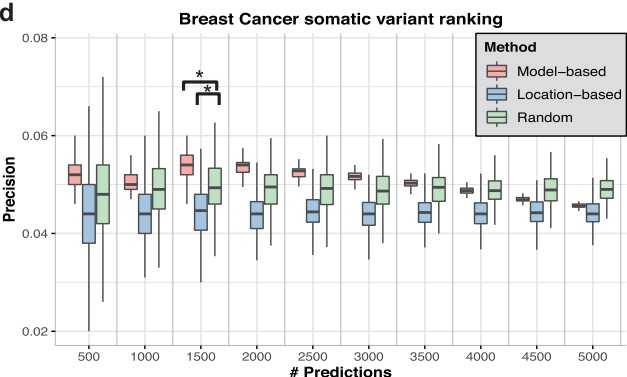

**d**

**e**

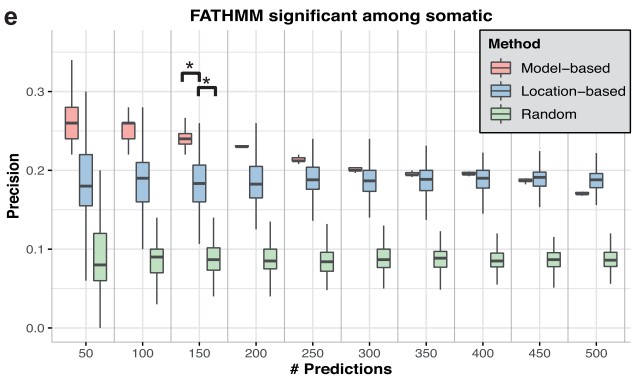

**Fig. 5 | GRN visualization and statistical assessment of model predictions. a** Venn diagram indicates the overlap between experimentally observed targets of AP2-γ (enhancers that are transcribed in estrogen-treated MCF7 cells but not so in AP2-γ knock-down condition) and model-predicted targets (enhancers whose predicted activity in AP2-γ knock-down is at least 5 percentile points lower). The overlap is statistically significant with Hypergeometric test *p* value 3e−4. **b** Visualization of the GRN components under the effect of ERα and AP2-γ cooperativity. The three layers of the network from bottom to top correspond to TFs, enhancers and genes, respectively. **c–e** illustrate evaluations of the variant impact prediction exercise. Three different methods for prioritizing functional variants were evaluated: an ensemble of GEMSTAT models used to predict variant impact on enhancer activity ("Model-based"), selection based solely on presence of ERα and H3K27Ac ChIP peaks ("Location-based"), and random selection ("Random"). Each method was used to prioritize a large number of common SNPs within enhancers from both classes, and top predictions were examined for presence of known breast cancer/tissue-related variants. *Y*-axis represents the proportion of True Positives among the prioritized variants and *X*-axis shows the number of prioritized variants, by each method. Known variants were defined to be breast cancer/tissue-related eQTLs, from GTEx and PanCan (**c**), non-coding somatic variants in breast cancer, from COSMIC (**d**), and FATHMM functionally significant somatic variants (FATHMM non-coding score >0.7) in breast cancer, from COSMIC (**e**) respectively.

control conditions to designate target enhancers for assessment of our predictions. Among the 2118 enhancers considered in our analysis, 720 enhancers were observed to be transcribed as eRNA in the control condition and 193 of those fail to transcribe in AP2-γ knock-down conditions and may thus be deemed AP2-γ targets (direct as well as indirect). The model-predicted regulon of AP2-γ includes 60 of the 720 enhancers, of which 29 were among the observed targets (*p* value = 3e−4, Hypergeometric test, Fig. 5a). Note that this significant result is based on predictions made from sequence analysis (GEMSTAT) alone, without consideration of experimental data on AP2-γ binding. The 60 putative AP2-γ targets include an unknown number of indirect targets of this TF, making the ~50% precision (29/60) a lower bound, which in our experience is a high level of agreement

between predicted and real TF targets[45]. Also, the target prediction as tested here goes beyond use of the general enhancer mark H3K27Ac or the specific mark of this system—ERα ChIP peaks, since these marks together define the "universe set" of enhancers evaluated. This result and the literature evidence for inferred TF roles together assure us regarding the biological and functional relevance of GEMSTAT models and their predictions.

Having derived an enhancer regulatory network, we next used genomic proximity (±10 kb) and chromatin interaction data on MCF7 cell line (obtained from 4Dgenome database[46]) to connect enhancers to genes. Putting together the two pieces of the puzzle, we obtained a breast cancer-specific GRN that connects TFs to genes through mediating enhancers (Supplementary Fig. 6). Such networks are of great interest[47] as they can

provide mechanistic hypotheses explaining the effects of cis and trans variations on gene expression. Another advantage of our approach to GRN construction is that it allows us to isolate enhancers that are under the effect of identified TF-TF interactions. As an example, Fig. 5b illustrates the GRN components affected by the cooperativity between ERα and AP2-γ. Enhancers included in this network are[1] predicted to be significantly affected by removing the specified interaction term between ERα and AP2-γ and[2] are experimentally observed to cease expressing (as eRNA) upon knock-down of AP2-γ.

### Mechanism-aware variant impact prediction

A major advantage of the trained ensemble of models is our resulting ability to predict the impact of genetic variations in regulatory DNA, by assessing the change in predicted expression due to a polymorphism. Importantly, predictions of high-impact variations in this approach are tied to a mechanistic explanation such as change in regulatory influence of a particular TF. To objectively test the proposed approach to impact prediction, we asked how well the models can prioritize breast cancer- and breast tissue-related variants among a large collection of common variants.

We prioritized variants from a large collection of over 36 million common variants and eQTLs. Out of about 36 million total relevant variants, 97,529 variants were located inside the 2118 enhancers considered in this study (described in Supplementary Data 1). We investigated the predicted effect for each of the 97,529 variants on the activity of their host enhancer. To do this, using each of the 244 GEMSTAT models in the final ensemble, we predicted the expression driven by mutant enhancers incorporating each of the variants. Using each GEMSTAT model, we quantified the predicted effect of the variant by comparing the predicted activity of the mutant enhancer (incorporating the variant) to its WT version. We finally short-listed ("prioritized") the top (say $K = 500, 1000, \ldots$, or 5000) variants ranked by predicted impact (i.e., model-based variants). These variants were then evaluated by their overlap with a "true set" of 1214 regulatory variants comprising breast-related eQTLs from GTEx and breast cancer-related eQTLs from the PancanQTL database, by noting down the fraction of prioritized variants that belong to the true set. We call this fraction the "precision" of the prioritized set. An example may help to clarify the precision calculations. For $K$ equal to 500, we evaluate the precision of the variant set as the number of variants in the set that overlap with 1214 known functionally relevant variants divided by the size of the set (i.e., 500). Repeating this procedure with each of the 244 GEMSTAT models in the final ensemble, we obtained as many estimates of precision of this model-based variant prioritization approach, shown in each red box of Fig. 5c (for different values of $K$).To assess the significance of model-based variant scoring, we then performed the same evaluations with a "location-based" approach where $K$ variants located within the enhancers were selected at random rather than based on predicted impact, thus utilizing functional information about enhancer locations but not the extra information provided by models of those enhancers. Again, by repeating the random sampling 244 times we obtained as many estimates of precision of this simpler strategy (shown as blue boxes in Fig. 5b). Finally, as a baseline approach, we selected $K$ variants at random from the entire collection, without regard to location within enhancers or model-based predictions, and obtained a distribution over their precision score (by repeating the sampling 244 times). These are shown as green boxes in Fig. 5c.

Evidently, the location-based approach consistently yields higher precision values than the random baseline for all examined values of $K$. This is unsurprising, as the enhancers used to filter variants in the location-based approach are ERα bound regions with potential regulatory activity, and are expected to be enriched for eQTLs. Importantly, the model-based approach yielded higher precision values than the location-based approach for a broad range of $K$ values (1500 … 4000). At $K = 1500$, for instance, this difference in precision is statistically significant with a $t$-test $p$ value of 2.02e−43. The two strategies are statistically indistinguishable for $K = 4500$ or 5000 and the model-based approach shows worse precision only at the lowest values of $K$ (=500, 1000). Considering a line connecting the median points of boxes

(Fig. 5c) of a strategy as a form of the popular "precision-recall" (PR) curve, one sees that the area under the PR curve is substantially greater for the model-based variant prioritization than the simpler, location-based method. This provides further evidence in favor of the underlying sequence-to-expression prediction models. Note that our goal here was not to present a variant impact prediction approach that improves on existing tools such as CADD[48], but to show by comparison between the model-based approach and the location-based approach that the model has learnt useful features of the underlying cis-regulatory logic.

We repeated the above evaluation using a set of 1438 breast cancer-related, non-coding somatic variants from the COSMIC database as the true set (Fig. 5d). We first noted, surprisingly, that the location-based strategy consistently shows lower precision than the random baseline, indicating that the somatic variants are significantly under-represented in the enhancers analyzed here. This observation may be explained by previous reports of reduced density of somatic mutations in areas of active enhancers[49,50]. On the other hand, once again model-based variant prioritization showed significantly higher precision than both baselines (location-based and random) for $K = 1500 … 4000$, and better than the location-based method for all values of $K$. Its precision was, however, worse than the random baseline at $K > 4000$, thus suggesting a range over predicted impact scores beyond which its advantage is lost.

A potential concern regarding the previous evaluation is that the breast cancer-related somatic variants may not be reliable as a "true set" of functional non-coding variants. To address this, we defined a smaller true set by utilizing the FATHMM[51] scores of variants provided by the COSMIC dataset[52]. FATHMM is a variant functional effect predictor that employs a wide range of features, including conservation, chromatin marks, accessibility and TF binding datasets, to predict function (pathogenic vs. non-pathogenic) of somatic variants. We thus used the set of 247 non-coding somatic variants with FATHMM scores higher than 0.7 as the true set; the full collection of variants from which the selection of top $K$ were made was changed accordingly to be all 1315 non-coding variants for which a FATHMM score is available. The model-based, location-based and random approaches for variant prioritization were evaluated as above, with $K$ set to 50, 100, … 500, to reflect the smaller collection. Results of this evaluation, shown in Fig. 5e, clearly indicate that functional non-coding variants are significantly enriched in the analyzed enhancers (blue versus green boxes) and that model-based impact prediction further improves detection of such functional variants significantly ($t$-test $p$ value < 4.7e−23 for $K = 50$, 100, … 300).

In summary, systematic evaluations shown in Fig. 5c–e demonstrate that by predicting the regulatory impact of non-coding variants using the ensemble of GEMSTAT models we can prioritize functional variants more effectively than by relying on epigenomic (ERα binding and H3K27Ac) data alone. In our view, this provides further evidence in support of the ensemble's ability to predict expression from sequence.

### Experimental validation of predicted variant impact mechanisms

Having built and tested a mechanistic model of the sequence-function relationship of ERα-responsive sequences that is also capable of predicting regulatory variant impact, we proceeded to use it to investigate three specific non-coding variants potentially relevant to breast cancer. These variants were selected based on their annotation as eQTLs or somatic variants in breast cancer, their proximity to putative target genes, predicted regulatory impact based on the GEMSTAT ensemble as well as experimental TF binding data supporting the model-suggested mechanism. Below we introduce each of the three variants, suggest a potential mechanism underlying its effect, and then report on experiments performed to test the proposed mechanism.

Shown in Fig. 6a, rs3809260 is a breast cancer (Pancan) and breast tissue (GTEx) related eQTL located on chr12 at 101697239 (hg38) and is characterized by reference/alternative alleles G/T. This variant is in the promoter of *CHPT1* gene and closer than 1 kb from its Transcription Start Site (TSS). The GEMSTAT ensemble predicts a 14 percentile increase in

activity (on average) of the enhancer carrying the alternative allele. (The WT enhancer has ERα and H3K27Ac ChIP peaks, but is not transcribed upon estrogen treatment.) The predicted change of expression is in agreement with Pancan data on breast cancer patients that show the alternative allele of this variant to be associated with higher expression levels of *CHPT1* gene (source: PancanQTL[53], beta = 0.17; see Supplementary Fig. 7). We suggest the following mechanistic explanation for the increased activity of *CHPT1* gene in individuals carrying the alternative allele: The mutation from "G" to "T" creates a putative FOXA1 binding site with relatively high predicted binding affinity (Fig. 6a, b). As evidenced by trained model parameters (Fig. 3f), and in silico knock-down experiments (Fig. 4e), FOXA1 is learned as an activator by our models. Moreover, the presence of FOXA1 in close proximity of this region is supported by multiple ChIP-seq experiments on the MCF7 cell line[44,54]. Increased affinity of DNA to locally present FOXA1 leads to increased probability of binding, leading to (additional) FOXA1-mediated activation of *CHPT1* gene for individuals carrying the alternative allele.

To test the above mechanism, we designed a variant of the *CHPT1* promoter where three nucleotide positions, including the SNP position, are mutagenized to create a consensus binding site for FOXA1 (see Fig. 6b and "Methods"), and measured its activity compared to the WT promoter in a dual-luciferase reporter assay (see "Methods" for variants sequences). As illustrated in Fig. 6c, experimental results showed that the variant with planted consensus FOXA1 site drives higher expression as compared to WT, both before and after treating MCF7 cells with estrogen-like agents (i.e., E2). This observation, combined with decreased activity of both WT and variant enhancers after E2 treatment, shows that the introduced FOXA1 site leads to increased baseline activity of this regulatory element and does not affect the estrogen-response dynamics. These findings support our hypothesis that stronger FOXA1 binding at the SNP's location causes increased promoter activity, which underlies the increased *CHPT1* expression associated with the alternative allele of the SNP. It is important to note that *CHPT1* gene is a direct target of estrogen (ERα binds its promoter upon estrogen treatment) and its inhibition is linked to decreased growth and cell proliferation in breast cancer cells[55].

The next variant was designed to probe the effect of the SNP rs12890411 on the activity of an ERα-bound enhancer region located in intron 5 of the *NOXRED1* gene. Illustrated in Fig. 6d, rs12890411 is a breast tissue GTEX eQTL as well as Pancan eQTL for HNSC, KIRC, LGG, and THCA cancer patients, whose allele is statistically associated with the expression of *NOXRED1* gene. The GEMSTAT ensemble predicts a 20 percentile lower activity (on average) for the enhancer harboring the alternative allele at this position. This prediction is in line with Pancan data from HNSC patients, which show lowered expression of *NOXRED1* gene in individuals carrying the alternative allele (beta = −0.28, Supplementary Fig. 8). As shown in Fig. 6d, e, G to A substitution at this position diminishes the predicted strength of RELA (NFκb) binding. (The RELA protein is known to bind to the WT enhancer, per ChIP-seq data[56]) We therefore suggest that the decreased binding affinity for this activator is at least partly responsible for the reduced expression of *NOXRED1* gene in individuals carrying the alternative allele. In order to test this hypothesis, we compared the activity of the WT sequence of the *NOXRED1* intronic enhancer (chr14:77399430-77400995) with the activity of a variant where the putative RELA binding site harboring the SNP is mutagenized (see Fig. 6e and "Methods"), using a reporter assay. Confirming the model predictions, experimental results in Fig. 6f showed a clear decrease in expression driven by the variant as compared to the WT enhancer before and after treating cells with estradiol (E2). It is evident from the increased reporter activity in E2-treated versus untreated conditions that this enhancer is responsive to estrogen. Interestingly, removal of the NFκb site did not hamper the response to estrogen, as both WT and variant enhancers responded similarly to estradiol (E2) treatment.

The third variant was designed to examine the effect of a somatic variant on the activity of an enhancer region located in the first intron of the *KEAP1* gene. The variant, called "Somatic_chr19_10502235_G_C"

(Fig. 6g), is a breast cancer somatic variant obtained from the COSMIC[52] dataset. The resulting variant enhancer is predicted by GEMSTAT ensemble to drive 17 percentile lower expression (on average) as compared to the WT enhancer. We note that *KEAP1* is a tumor suppressor gene known to be down-regulated in breast cancer, and *KEAP1* promoter methylation is specifically pronounced in ER+ breast cancer patients[57,58]. As shown in Fig. 6g, h, the somatic variant is predicted to weaken a binding site for NR5A2, a TF learned by the model to be an activator of enhancer activity. (This TF has a ChIP-seq peak encompassing the variant position[59]) Hence, we propose the diminished DNA-binding of NR5A2 to cause the reduced expression of *KEAP1* in breast cancer patients carrying this somatic mutation. To test our hypothesis, we compared the activity driven by the WT enhancer to the activity of a variant in which the predicted NR5A2 binding site harboring the somatic variant is mutagenized (see Fig. 6h and "Methods" for sequence), through dual-luciferase reporter assays. Experimental results reported in Fig. 6i illustrated that the variant enhancer drove lower expression (i.e., normalized luciferase activity) as compared to the WT in both pre- and post-E2 treatment conditions. Moreover, these results suggest that removal of NR5A2 site alters the estrogen-response dynamics. This is evident from the observation that, as opposed to the WT enhancer, the variant enhancer showed no increased expression upon E2 treatment.

## Discussion

Sequence-based gene expression prediction has proven to be a challenging task, especially in mammalian systems where the regulatory sequence controlling a gene can be located virtually anywhere on the genome. Several technological and analytical advances have pushed the frontiers in this area. Epigenomic profiles provide valuable clues regarding the locations of active enhancers[60], sequence-to-expression modeling has been successful for individual enhancers[25], and enhancer-promoter interactions revealed by chromatin conformation data[61] help assign enhancers to their target genes. Yet, quantitative mapping from sequence to expression remains elusive. One reason for this is our poor understanding how the readouts of multiple enhancers combine to yield the gene's overall expression level. There has been some work toward this, e.g., where enhancer outputs are combined via a weighted summation scheme[26,62], but the weights are generally unknown and the challenge of simultaneously modeling outputs of individual enhancers and combining them to predict gene expression remains daunting. In this context, modeling eRNA transcription provides a useful workaround. Here, we only need to model the eRNA transcript level as a function of its own sequence. At the same time, the strong correlation between eRNA levels and associated gene expression levels[16] means that modeling eRNA sequence-to-function relationship can provide valuable insights into cis-regulatory logic of a gene. This insight is central to the methodology we adopted in this work.

We framed the broader challenge of characterizing the ERα transcriptional program in a more specific manner: to quantify the sequence encoding of eRNA transcription observed during this extensive transcriptomic response. We first systematically narrowed down the set of potential ERα coregulators, curated manually from the literature, to the ones that best distinguish transcribing from silent enhancers, using standard machine learning methods. In order to shed light on mechanistic functions of these short-listed regulators, we then placed them in a thermodynamics-based framework (GEMSTAT) and trained its biophysically interpretable parameters on the eRNA data. Roles learned by an ensemble of GEMSTAT models for the majority of studied regulators were found to be in agreement with previous observations in the context of ERα regulatory program.

In recent years, promising progress has been made in sequence-to-function modeling, where function includes epigenomic states, TF binding, etc. Some of these supervised learning-based approaches are not mechanistically grounded but choose to investigate mechanisms in post-processing steps[63–65]. Other mechanism-based sequence-to-function models[66] predict the binding of individual TFs but do not go beyond that to consider the collective impact of multiple TFs binding to the same enhancer. GEMSTAT provides a mechanism-aware framework to study the sequence-dependent

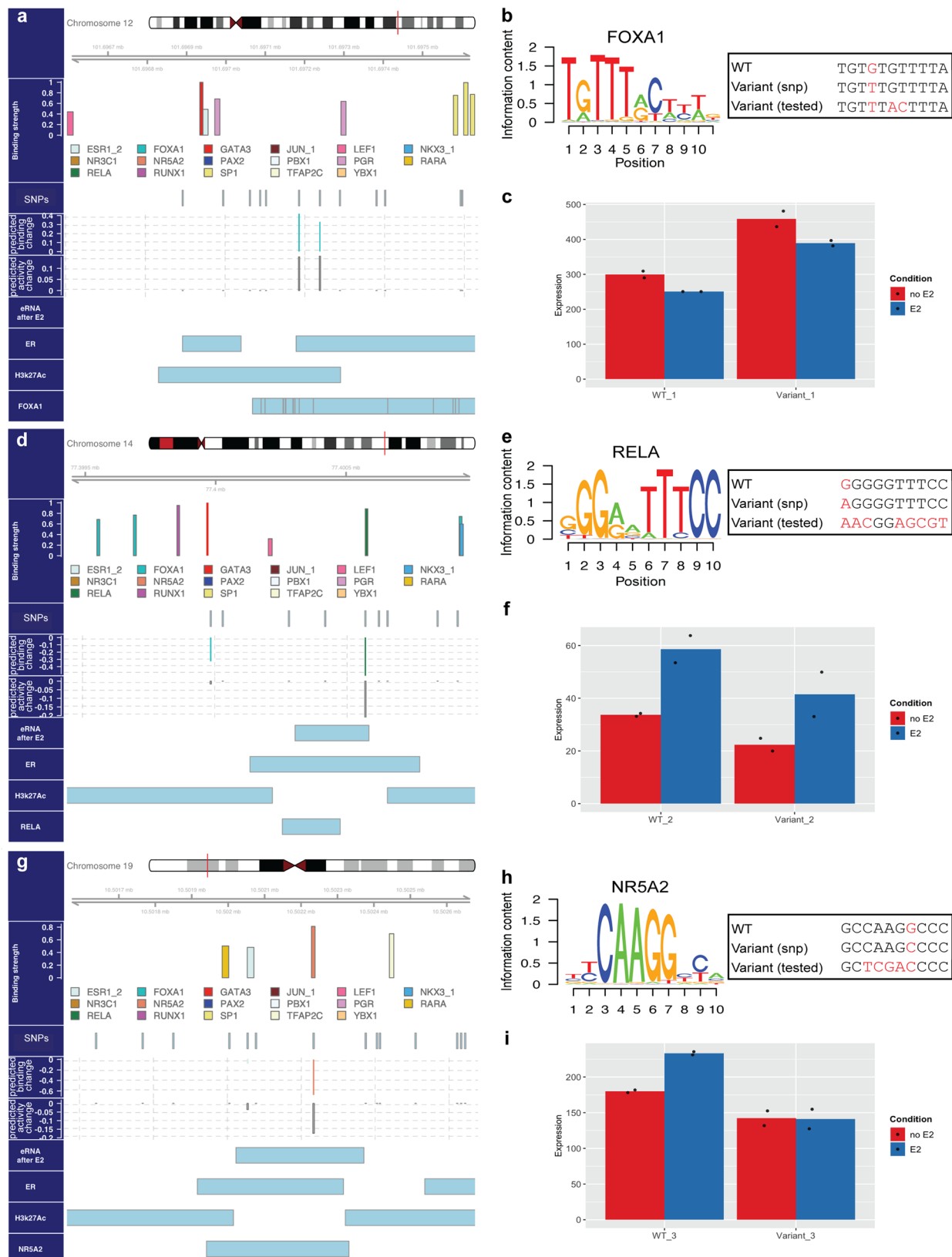

combinatorial action of TFs and investigate the rules that determine the resulting transcriptional activity.

Trained sequence-to-expression models are capable of predicting the TF-mediated impact of sequence variants on enhancer function[66,67]. Since our models were trained on data representing the ERα transcriptional program, and aberrations in this program are associated with breast cancer[3], we expected high-impact variant predictions from our models to be enriched for breast cancer relevance. This was indeed the case in our systematic evaluations, adding support to the reliability of the models. Notably, the extent of enrichment was greater when using our models for variant

**Fig. 6 | Selected breast cancer-related SNPs with potential regulatory function.** **a**, **d**, **g** show evidence for our proposed hypothesis explaining the underlying mechanism of action for the studied variants. Each panel contains multiple tracks characterizing the region harboring the variant. From top to bottom, horizontal tracks depict the chromosomal location of the 1 kb region, color coded motif-based estimate of binding strength for all TFs included in the models (Binding strength), locations of common SNPs, breast tissue/cancer eQTLs and somatic variants (Variants) present in the vicinity of the studied variant highlighted in gray, predicted change in binding strength due to a variant, measured by the change in LLR score of a binding site normalized by maximum LLR score among all sites of that TF (Binding change), predicted percentile change in enhancer activity due to a variant (Activity change), presence or absence of transcribed eRNA after E2 stimulation of MCF7 cells (eRNA after E2), and presence or absence of ERα, H3K27Ac and other TF ChIP peaks. **a** Schematic representation of the enhancer region containing rs3809260 variant (highlighted). **b** Tested variant enhancer differs from wild-type enhancer in a putative FOXA1 binding site sequence; three nucleotide changes (including the SNP) create a consensus FOXA1 site. **c** Results of dual-luciferase assays on WT enhancer (shown in **a**) and variant (tested) enhancer, in MCF7 cells without or with E2 treatment. Note that experiments were done in duplicates. **d** Schematic representation of the enhancer region containing rs12890411 variant (highlighted). **e** Tested variant enhancer differs from wild-type in a putative RELA binding site sequence; a strong RELA site is abolished in the variant. **f** Results of dual-luciferase assays on WT enhancer (shown in **d**) and variant (tested) enhancer, with and without E2 treatment. **g** Schematic representation of the enhancer region harboring the somatic variant chr19_10502235_G_C (highlighted). **h** Change in putative NR5A2 binding site sequence in the tested variant enhancer; a strong NR5A2 site is abolished. **i** Results of dual-luciferase assays on WT enhancer (shown in **g**) and variant (tested) enhancer, in MCF7 cells with and without E2 treatment.

prediction compared to relying solely on epigenomic information. At the same time, we did not expect our models to identify every breast cancer/tissue-related sequence variant as high-impact, for two reasons. First, our model-based approach can only identify the TF-mediated changes in expression due to a variant. Hence, the trained models will not prioritize variants that act through other mechanisms such as lncRNA mediated changes in transcription. Second, many of the variants annotated by previous studies as breast tissue/cancer-related are merely in linkage disequilibrium with casual variants and are not expected to drive changes in expression. Our model-based variant scoring is not expected to prioritize such variants. Due to these two reasons, the assignment of true positive and true negative values in our evaluations of variant impact prediction are unreliable in absolute terms, but useful for comparison of alternative variant prioritization strategies based on relative accuracy.

A key advantage of our approach is to provide testable mechanistic explanation for the predicted impact of a sequence variant, as being the result of changes in TF binding strength. Note that an increase in TF binding strength may translate to increased or decreased predicted enhancer activity depending on the TF's role, which is also learned by the models. To highlight the mechanism-aware variant impact prediction capability of the models, we tested three high-impact variants through reporter assays. We devised minimal experiments to provide evidence for the proposed hypothesis explaining the observed alteration in expression of the gene presumably controlled by each considered enhancer. In each test case, our models suggested that the considered mutation leads to a change in binding affinity of enhancer to a particular TF, causing the observed transcriptional change. Hence, we tested the reporter activity of enhancer constructs that were predicted to result in the utmost departure from WT enhancer activity in the same direction as that of the considered mutants (i.e., increased/decreased activity for eQTLs with positive/negative coefficient, respectively), by completely abolishing or maximally strengthening the binding site for the TF identified to be responsible. Experimental observations found model predictions to be qualitatively correct in all three cases, providing evidence for the role of FOXA1, NFκb, and NR5A2 in mediating the effect of functionally significant studied variants. In order to provide further evidence for our hypotheses, follow-up experiments can be performed to quantify the change of DNA binding of the suggested TFs, as resulted by the sequence variants.

One of our analyses (Fig. 5d) found that breast cancer somatic mutations are less frequent in the ERα-bound enhancers compared to random expectation. At the first glance this result was surprising. Functional variants, e.g., eQTLs, have been reported to be enriched in relevant enhancers[68]. Moreover, one of the main characteristics of cancer is increased rate of mutation, and these numerous somatic mutations may be naïvely expected to localize in enhancers and other open chromatin regions that are more exposed to mutation causing agents[69]. However, this intuition seems to be incorrect, as transcription-coupled DNA repair mechanisms protect open DNA regions against point mutations[49]. Studies have found lower levels of point mutations in areas of open chromatin. A recent study[50] found that, as opposed to

copy number aberrations (CNA), somatic point mutations have a slight negative correlation with enhancer activity across various cancer types. This finding appears consistent with our observation of breast cancer-related somatic mutations being depleted in ERα-bound enhancers.

To the best of our knowledge, this is the first thermodynamics-based sequence-to-expression model successfully applied to a large-scale mammalian system. A hallmark of such models is their ability to combine enhancer sequence and cellular TF concentrations to predict condition-specific expression. However, due to the lack of comprehensive data on TF concentration measurements in the E2-treated MCF7 cell lines, our models currently examine only sequence differences among enhancers and are limited to making predictions in conditions similar to the training condition. We should also note that GEMSTAT requires a set of defined TF motifs and does not perform de novo motif identification. Despite these limitations, GEMSTAT offers a way to interpret enhancer sequences while accommodating sequence properties relevant to function, e.g., numbers and strengths of binding sites, overlaps between sites, cooperativity, and antagonism between TFs. Future extensions of our work will address this modeling limitation, e.g., by inference of effective TF concentrations from available ChIP-seq data.

## Conclusions
Here, we decipher the cis-regulatory logic of ERα-mediated transcriptional response to estrogen in breast cancer cells, through an integrative approach that leverages the power of multi-omics data, and benefits from combining sequence analysis and mechanistically grounded biophysical models. We demonstrated that this approach produces testable mechanistic hypotheses, identifying TFs regulating a given enhancer and shedding light on the chain of events triggered by sequence-mediated aberrations that lead to the malignant transformation of normal cells. Previously reported experimental results, as well as our own experimental observations confirm the sequence-based model's predictions, providing support for its validity.

Based on in silico knockdown analysis, we concluded that the models learned an activating role for ERα, FOXA1, GATA3, NR5A2, PAX2, PBX1, and RARα. This activating role combines the binding strength and activation potential for each given TF. In case of ERα most of the models learned a strong activation parameter paired with a weak binding parameter. We also noted that the models learned strong cooperativity parameters between ERα and several TFs including FOXA1, GATA3, and RARα, all of which are leaned as strong binders. The high activation parameter, and weak binding parameter learned for ERα, together with its strong cooperativity suggest a potential mechanism in which ERα possesses strong transcriptional activation potential but a weak DNA binding capability, and thus utilizes co-factors such as FOXA1 and GATA3 for DNA binding. Additionally, most models assign a repressive role to Glucocorticoid Receptor (GR). Similarly, almost all models learn a strong cooperative interaction between ERα and RARα. The interplay between ERα, YBX1, and RUNX1 can be an interesting future direction to explore.

## Methods

### Classification group setup

We considered the full set of 3734 ERα binding sites with H3K27Ac marks in estrogen-treated MCF7 cell line (Li et al.[10]) as the "universe set" of genomic segments to analyze. To identify 3734 potentially active enhancers, we used H3K27Ac and ER ChIP-seq processed data (in bigwig format) published by the authors. More specifically, In each processed dataset, we used a threshold score of 5 on peak scores and merged peaks within 100 bp of each other. Note that our criteria to identify enhancers is different from that used by Li et al In that study authors also considered H3K4me1 peaks to mark potential enhancers, whereas we took the intersection between the peaks found in H3K27Ac and ER ChIP-seq datasets to get to the set of 3734 potential enhancers. Li et al. also conducted GRO-Seq experiments before and after estrogen treatment (two replicates per condition) to identify actively transcribed regions. We used these data to define positive and negative segments among the universe set. The positive set includes 449 segments that overlap with at least one GRO-Seq peak in both post-treatment replicates but do not overlap with peaks in either pre-treatment replicates. (We excluded 40 out of 489 eRNA-overlapping examples shown in Fig. 2a because of overlap with pre-treatment GRO-Seq peaks.) The negative set includes 1669 genomic segments that lack the eRNA signal in both of the post-treatment replicates. (We excluded 1576 out of 3245 non-eRNA-overlapping examples shown in Fig. 2a because of overlap with GRO-Seq peaks in one of the replicates.) The above segments were randomly partitioned into training and test sets that comprised 1336 and 333 negative examples, complemented with 360 and 89 positive examples, respectively.

### Feature set construction

We conducted a comprehensive literature review in order to identify a set of 30 TFs that have been suggested to play a role in the ERα transcriptional program (Supplementary Table 3). Thirty three PWMs for these 30 TFs were downloaded from CIS-BP database[70]. Note that we used three and two motifs to represent ESR1 (ERα) and JUN TFs, respectively. Each enhancer was scanned using a TF's PWM to identify and quantify putative binding sites, and likelihood ratio (LR) scores of all putative sites were summed to obtain a single measure of the enhancer's affinity for the TF. In order to construct features capturing pairwise TF interactions, we set an LR score threshold for each TF corresponding to a $p$ value of $1e-4$, and marked putative sites stronger than this threshold. Next, for all pairwise combinations (total of 561 pairs) of the 33 used motifs, including homotypic-pairs, we counted the number of adjacent sites (within 50 bp of each other) in an enhancer; this count was used as the pairwise adjacency feature of the TFs. In summary, we used 33 features representing individual motif scores, together with 561 features representing their pairwise adjacency profile (i.e., total of 594 features) to train the random forest classifier.

### Random Forest classification

Random Forest classification was done using caret package[71] in R. Down-sampling was performed during training in order to deal with the class imbalance problem. The number of variables available for splitting at each tree node (mtry = 66) was chosen using repeated 6-fold cross-validation. There were 1000 classification trees in the Random Forest model, each containing on average 167 nodes. ROC and PRC were calculated and drawn using PRROC and ROCR packages[72,73] in R. The importance of each feature in the Random Forest was calculated using the average change in "out of bag error" of all trees in the RF upon permuting that feature.

### GEMSTAT modeling

GEMSTAT[24] is a thermodynamics-based model that uses the sequence of an enhancer, PWMs of the TFs that regulate the enhancer and concentration of those TFs to predict the expression driven by the enhancer. It estimates the energy of TF-site interactions based on the well-established relationship between PWMs and binding energies[74], and uses these energies along with terms (free parameters) representing protein-protein interactions to define a Boltzmann distribution over all possible configurations of bound and unbound sites in the enhancer and bound or unbound Basal Transcriptional Machinery (BTM) at the promoter. The enhancer's expression readout is then assumed to be proportional to the fractional occupancy of the BTM, calculated by averaging over this distribution. The Boltzmann weight of each configuration is calculated based on affinity of occupied TF binding sites (DNA-protein interaction) and interaction between bound TFs (protein-protein interaction) in that configuration, as well as cellular TF concentrations. The interaction of a bound BTM with each of the bound TFs may increase or decrease the weight of that configuration depending on the TF being an activator or a repressor. GEMSTAT predicts the relative expression driven by an enhancer as the ratio of total weight of configurations where BTM is bound to the total weight of all configurations.

In this study, we modified GEMSTAT to minimize the following objective function during parameter training, in order make it suitable for a binary classification task:

$$\sum_i \ln\left(1 + e^{-1 \times y_i \times \alpha \times (\widehat{y_i} - \beta)}\right)$$

where

$y_i \in \{-1, 1\}$ is the label for enhancer $i$; $\hat{y}_i$ is the predicted (numeric) activity for enhancer $i$; $\alpha$ is the logistic coefficient learned as a free parameter, $\beta$ is the bias free parameter. There are a total of 44 free parameters learned by GEMSTAT. To create an ensemble of GEMSTAT models (where each model is an assignment of values to free parameters), we initialized the models with distinct parameter settings, obtained by uniform sampling from the parameter space[27]. In the final ensemble, we included an additional set of initial points comprising learned parameters from previous rounds of optimization (see Supplementary Note 2).

### Variant impact quantification

The predicted effect of an individual variant on a single enhancer's activity is computed as the difference between (percentile normalized) predicted activity of mutant and WT versions of that enhancer. In particular, for each GEMSTAT model, we constructed the distribution of predicted activities of all WT enhancers (positive and negative classes) and used it to assign a percentile score to any predicted activity value. The percentile normalized activity of the WT enhancer and the mutant version due to a variant were compared and their difference was deemed as the effect of the variant on the enhancer's activity, according to the particular GEMSTAT model. The ensemble of models yields a corresponding ensemble of predicted effects for any variant.

Since GEMSTAT is designed to maximize the separation between the two classes of enhancers by performing a binary logistic regression classification, the absolute value of predicted expression is not directly interpretable. However, the relative placement of the predicted expression value for an enhancer in the distribution of such values for all enhancers—captured by the percentile measure—better reflects the intended use of the optimization results.

### Data processing

Processing of bed files was done using BEDTools[75]. Further data processing and statistical tests were done using R programming language[76] version 3.4.2. Reading, processing, and plotting of ChIP-Seq data were done using ChIPseeker package[77]. GRN Visualization was done using mully package[78] in R, and enhancer genome track visualization was done using Gviz package[79] in R.

## Final tested variant sequences

The altered segment of the *CHPT1* promoter (predicted FOXA1 binding site) used in the first variant is described below:

```
WT:          TGTGTGTTTTA

variant:     TGTTTACTTTA
```

The sequence used in place of RELA binding site to construct the second variant:

```
WT:          GGGGGTTTCC

variant:     AACGGAGCGT
```

Finally, we describe the changes made to the WT enhancer associated with *KEAP1* to remove the NR5A2 binding site (third variant):

```
WT:          GCCAAGGCCC

variant:     GCTCGACCCC
```

The full sequence for all six tested WT and variant regulatory elements, as well as their associated constructs can be found in Supplementary Data 3.

## Cell culture

MCF7 cells were cultured in RPMI supplemented with 5% FBS in a humidified incubator containing 5% $CO_2$ at 37 °C. The cells were hormone-stripped for 72 h in phenol red-free RPMI containing 10% charcoal-stripped FBS prior to estradiol (E2) induction.

## Luciferase assay

The wildtype and variant enhancer regions were cloned into pGL4.27 firefly luciferase vectors (Promega) by GenScript. The firefly luciferase-containing constructs were co-transfected with pGL4.74 Renilla vector (Promega) using Lipofectamine 3000 (Invitrogen). Five hours after transfection, the cells were treated with 10 nM estradiol (E2, Sigma-Millipore) or ethanol (no E2 control) for 24 h. After E2 or no E2 treatment, the cells were rinsed with PBS then lysed with Passive Lysis Buffer (Promega). The Dual-Luciferase Reporter Assay System (Promega) was used according to the manufacturer protocol. Readings were measured in duplicates using the Ultra Evolution Microplate Reader (Tecan). Each sample was prepared in duplicates and the luciferase activities were normalized by dividing firefly by Renilla to control for transfection efficiency.

## Data availability

ERα and H3K27Ac ChIP-Seq datasets were downloaded from GEO database with accession number GSE45822. All GRO-Seq data (on E2-treated MCF7 cells[10], and E2-treated and control AP2-γ knock-down MCF7 cells[44]) were obtained them from Nascent Transcription Repository[80]. Genetic variants used in this study were obtained from the sources reported in Supplementary Note 1. Common SNPs were obtained from dbSNP build 151, subject to a minor allele frequency threshold of greater than 1%. Processed and raw data from luciferase assays are provided as Supplementary Data 4 and 5, respectively. The source numeric data behind the graphs presented in this manuscript are provided as Supplementary Data 6.

## Code availability

GEMSTAT software version used in this study is available here: https://zenodo.org/badge/latestdoi/139994091; https://github.com/ShayanBordbar/GEMSTAT. Other scripts regarding RF models, constructing GEMSTAT ensemble inputs, analysis and visualization of outputs, and motif *p* value calculation (script motif_pvalue.pl developed by S.S.) are available here: https://github.com/ShayanBordbar/ER_scripts; https://zenodo.org/badge/latestdoi/307785178.

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

## Acknowledgements
This work was funded in part by the National Institutes of Health (grants R35GM131819 and R01GM114341A to S.S.). K.V.P. laboratory is supported by grants from Cancer center at Illinois seed grants and Prairie Dragon Paddlers, NSF EAGER MCB1723008, NIH R01GM132458 and R21AG065748. The MCF7 cells were a kind gift from Dr. Benita Katzenellenbogen (UIUC). We acknowledge Dr. Brian Freeman (UIUC) for his help with the luciferase assay.

## Author contributions
S.T.B. and S.S. designed the study. S.T.B., S.S., and Z.A. wrote the manuscript. S.T.B. performed data analyses and visualization. K.V.P. and Y.J.S. designed and performed dual-luciferase reporter assay and wrote the corresponding method sections. B.J.L. developed the initial version of GEMSTAT software that was further modified for this study by S.T.B.

## Competing interests
The authors declare no competing interests.
