## [Peer Review File · Communications Biology]

Reviewers' comments:

Reviewer #1 (Remarks to the Author):

In this manuscript, the authors report their computational analyses of estrogen receptor alpha (ERalpha)-bound enhancers. They chose ERalpha-bound enhancers using the ChIP-seq data of ERalpha and H3K27ac, and then divided them into positive and negative data based on their expression of enhancer RNAs. Combining with a set of known relevant TFBS information, they chose important sequence features using the random forest algorithm and they constructed simpler prediction models with fewer parameters. These models were further used for in-silico perturbation experiments and the prioritization of breast cancer-related sequence variants. Finally, the predicted effects of some variants were tested by the dual-luciferase reporter assay. Overall, I think that this is an interesting computational study though newly-obtained biological insights may not be so significant (in this sense, I feel that the latter part of their results could be shortened).

Here are some of the points I would like the authors to consider to improve the current manuscript:

1. Although they first attempted to build a thermodynamic model for discriminating eENA-producing and non-producing enhancers, the question on which are the main sequence differences were not explored well (nor the interpretation of their obtained thermodynamic model). Although it makes sense to show AUROC/AUPRC results, it may be biologically more important to show how much the two classes of enhancers are different in terms of the presence/absence of several key motifs.
2. The reason why they took two steps in model building is not clear. It seems more straight-forward to use, say, LASSO directly. If they claim that their method is better, they should show the result of their comparison.
3. In Table 1, they claim that they could confirm the significance of their results through literature survey. However, they used a restricted set of TFs from the beginning and thus it does not make much sense.
4. In my opinion, many of the figures are poorly presented:
 - i) Fig.2a is not referred to in the Results and it does not explain the subsets they used in their study.
 - ii) In Fig.3f, the differences between the three groups are hard to recognize.
 - iii) In Fig.4d/e, which motifs are significant is hard to recognize.
 - iv) Figs.6a/d/g are hard to interpret.
5. In the results presented in Figs. 5c/d, they used a location-based model as a control, how about using a simple weight matrix-based scores within the enhancer regions, instead? In other words, how much better is their model than a simple WM-based model based on detected significant motifs?
6. I wonder if their reporter assay experiments are supportive enough for their models. The results of changing the consensus sequence of known TFBSs sound somewhat trivial. How about, for example, changing the sequences of negative enhancers? In this sense, they should confirm if the negative enhancers are free from breast cancer-related variants.

Reviewer #2 (Remarks to the Author):

Brief summary of the manuscript:

Manuscript: COMMSBIO-22-1690

Title: Mechanistic analysis of enhancer sequences in the Estrogen Receptor transcriptional program
Authors: Tabe-Bordbar et al.

The transcription factor (TF) Estrogen receptor alpha (ER α) translocates into the nucleus upon binding estrogen analog (E2) and binds thousands of estrogen response elements (EREs) in the genome. The majority of EREs and ER α -binding events occur in gene-distal or intragenic regulatory sequences (enhancers), while a minority occur closer to gene promoters. Upon binding to the respective EREs, ER α nucleates formation of multiprotein complexes of other TFs and coregulators (CoRegs), which then make physical contacts with the protein complexes assembled at target gene promoters. These events instigate transcription both at the enhancers (producing eRNAs) and at the “in contact” target promoters (initiating synthesis of mRNA).

This manuscript relies on the pioneering work done in the lab of Geoff Rosenfeld in MCF7 cells (Li et al, Nature, 2013) that (1) of thousands of E2-induced ER α -binding events, a subset coincides with E2-induced enrichment of H3K4me1 and H3K27ac, thereby identifying these sequences as potential enhancers, and (2) these enhancers undergo transcription producing eRNAs. The paper goes on exploring further mechanistic questions in E2-induced transcriptional regulation that are not relevant here.

The current manuscript identified that of the E2-induced ER α -bound enhancers in the dataset of Li et al, only a subset produces eRNAs (called by the authors +ve enhancers) while a larger subset does not (called -ve enhancers); the rest of the enhancers are ignored. The authors were interested in understanding what makes only a subset of ER α -bound enhancers transcribe. They employed Random Forest analysis to classify the sequences of +ve and -ve enhancers and identified differing TF and TF-pair occupancies. Both classes of enhancers were then analyzed with a bioinformatic pipeline, GEMSTAT, that the lead author’s lab worked out in 2010 (He et al, PLoS Comp Biol). [GEMSTAT takes into account multiple criteria such as the TF binding sequences (TFBSs) and their thermodynamic properties dictating TF-DNA interactions, the position weight matrices (PWM) of the TFBSs, the TF availability in the cell, and assigns the enhancers a predictive value for expression.] To validate the predictive power of their GEMSTAT analysis, the authors then conduct in silico knock down of TFs. These analyses are extended to analyze disease specific sequence variants of enhancers in an attempt to understand their altered functions.

Overall impression of the work:

This is an interesting work. eRNAs have predictive gene-regulatory power, and there are efforts to target eRNA expression in breast cancer therapeutics. So, knowing what sequence features of enhancers dictate eRNA expression and how this is dysregulated in disease-specific variants will have immense value. However, even as they seem to have started the work with this goal, the authors seem to have gone astray a bit (details below). This manuscript would be apt for the journal; but there are several concerns in the manuscript that must be addressed before the manuscript can be considered for acceptance. The authors are a team of accomplished scientists and addressing these concerns should be easy.

Specific comments, with recommendations for addressing each comment:

The comments are listed in the order the concerns appear in the manuscript.

1. Page 2: in the “conclusions” section, the authors simply state ‘gained mechanistic insights... program’. Please specify the concluding points/outcomes (mechanistic insights gained) of the work here.

2. Page 7: This work is based on the ChIP-seq/GRO-seq data sets of Li et al. 2013. However, the numbers do not seem to match. Li et al identified 7174 potential enhancers enriched for ER α , H3K4me1 and H3K27ac upon E2 treatment; of these, 1045 occur within 200kb of a gene TSS upregulated by E2 (of these 1033 bind ER α only at the enhancers; see Fig S1). Thus, these 1145 were considered as E2-responsive enhancers.

a. So, where does the number 3734 as ‘potentially active enhancers’ come from? Do the authors use different thresholds to analyze sequence reads? This needs clarification.

b. Of the above 3734, the authors designated 449 as +ve (for E2-induced eRNA expression) and 1669 as -ve enhancers (no E2-induced eRNA), and the rest are ignored. This is a curious case. How do the authors quantify eRNA? Since (1) eRNAs are thought to mark “active” enhancers (that are implicated in activation of a target gene undergoing transcription), and (2) the magnitude and temporal kinetics of eRNA production correlate with the production of the target mRNAs (thus, the magnitude of eRNA expression can be reflective of an enhancer’s activity, as well as of the target gene’s transcription, and the authors seem to accept this assumption), the 3734 ‘potentially active enhancers’ are expected to (1) activate target gene(s) and (2) produce eRNAs. How do these enhancers contain 1669 that do not transcribe? If they do not transcribe eRNAs, are they still activating target genes? Or, are the authors hinting at a set of active enhancers activating target genes yet not transcribing themselves? That would be a paradigm-shifting notion. And, if so, do these enhancers engage in contact with the target ‘transcribing’ promoters while themselves not transcribing? Further, what are the “ignored” enhancers? Since all enhancers considered here are ER α -bound, they all can come in only two possible classes: either transcribing eRNAs or not transcribing eRNAs. What is this third class that was ignored? These questions are relevant and need answers since these form the very premise that the entire manuscript rests upon.

c. For the benefit of the readers and reviewers alike, I ask that the list of the coordinates of the +ve, -ve and “ignored” enhancers (and preferably their potential target genes) is provided as a supplemental file, along with their relative E2-induced sequence reads (or fold-change) of the eRNAs and corresponding target mRNAs.

3. Page 7/8: Figures 1 and 2a are not cited. Please make sure that all figures are cited in the main text in the order they are numbered.

4. Page 8, line 177: What is a ‘numeric feature’? How were 594 of them derived? This should be explained in the methods or in a supplemental text. At the end of this paragraph, the authors select 16 of these numeric features, which are listed as TFs in Table S1. So, is a TF considered a numeric feature? Why? What were the criteria to select “16 most important features” out of these? These need to be explained somewhere.

5. Page 8, line 178: How was the “adjacency score” calculated? No explanation anywhere. What does a -ve value mean (Fig 2b)?

6. Page 8, line 183: There is insufficient methods on how AUROC and AUPRC values were derived, and no explanation on how to interpret their numeric values. The authors should consider the fact that numerous researchers follow classical gene regulation and hormone regulation who may not necessarily have exposures to the authors’ line of computational approaches.

7. Page 9, line 203: Reference 25 is the same as reference 21. Please remove the duplicate.

8. Page 9, line 204: More description needed on GEMSTAT as adopted here. The current GEMSTAT analysis is based on the initial analytic pipeline in He et al, 2010, where mathematical modeling included thermodynamic properties of TFBSs, cellular abundance of TFs, known or predicted TF-TF and TF-basal transcription machinery interactions in *Drosophila*. The present scope of GEMSTAT seems much expansive compared to the *Drosophila* work, where only a handful TFs were considered. Plus, the current work appears heavily compromised as it does not consider TF availability and Pol II occupancy at the enhancers. Quantitative TF availability can be found elsewhere (for example, Nusinow et al. *Cell*, 180: 387; 2020), while the authors can conduct their own Pol II ChIP-seq if not available with Li et al, 2013. Another complexity could “potentially” compromise the predictive power: the TF occupancies at the promoters – both core promoter elements and proximal promoter TFBSs, plus Pol II (it would be naïve to think that the enhancer alone determines the transcriptional output of a gene, with no contribution from the promoter). While I understand that these expansions in the analytic framework may not be possible at the moment, it nevertheless makes this manuscript look undercooked.

9. Page 9, line 217: How are the “17 TFs and 9 TF-TF interactions” selected and why? Are these related to the 16 numeric features of Table S1?

10. Page 10; also Fig 3: What criteria were used to identify and differentiate the “binding” and “activation” parameters, and what does a -ve value mean here (Fig 3f)? This needs to be explained. Classically speaking, the binding and activation functions of a TF are tricky to separate. A TF may bind the target site strongly but may not strongly activate the gene, but a strong activator is “likely” to show stronger binding. This is where this analysis does not inspire confidence, because of conceptual inconsistency. Here are some examples:

- a. Despite the entire dataset based on ER α -bound enhancers for ER α -dependent transcriptional activation, Fig 3f shows no enrichment in ER α binding. How is it possible that analyses critically and centrally based on ER α -enriched datasets do not exhibit ER α enrichment?
- b. Despite being a strong activator (Fig 4c), PBX1 scores very poorly for activation in Fig 3f. Why this discrepancy in data?
- c. YBX1, a strong repressor (Fig 4c; Table S2) shows in strong binding in group 1 and 3 enhancers with less repression, yet binds feebly to group 2 enhancers with stronger repression (Fig 3f). This appears to defy logic.
- d. AP2- γ does not bind the enhancers (its binding score is -ve; but is there an alternative interpretation?), yet has moderate activation function – though is listed as a strong activator (Table 1).
- e. FOXA1 “activates a large fraction, nearly 50%, of examined enhancers” (page 12, line 291; and Fig 4),

and yet scores around zero for activation in Fig 3f.

These inconsistencies perhaps indicate that the predictive interpretations of various analytic readouts lack coherence. These inadequacies also point to the compromised nature of the work as pointed above.

11. Page 11: The validity, relevance and significance of the in silico perturbation experiments are unclear. Here is why. The RF and GEMSTAT analytic pipelines were trained for enrichments of certain TFs. So, when some of those TFs would be omitted in in silico perturbation, it is natural that the readouts will be adversely affected; there is no novelty here. For example, a dataset trained on an activator will predict loss of activation when the activator is omitted. Instead, the authors need to conduct real MCF7-based knockdown experiments and show that:

- a. A TF upon knockdown leads to loss of transcription from an enhancer it is predicted to activate
- b. In the same experiment, those enhancers are unaffected where the TF is not predicted to activate. I believe the authors' GEMSTAT and Regulon analyses can likely predict even secondary regulatory effects.

12. Page 15, lane 350 (also page 27, line 639): I strongly suggest omission of the term "universe". Scientific communications should not encourage scientifically incorrect symbolism. "Universe" essentially means "all that there is, was, will be – ever" and goes unfathomably beyond the "observable universe", which has a radius of over 46 billion light years. It is all-inclusive – known or unknown, with absolutely no exception. However, the usage "universe" of enhancers is so specifically exclusive minuscule an entity here that it does not even include 'all potential enhancers in human genome – active or otherwise', let alone all enhancers ever evolved in any organism. Such a narrow usage will be a disrespect to our universe.

13. Page 16: Mechanism-aware variant impact prediction:

a. It is not readily clear what is a "model-based variant" (red boxes in Fig 5c-e). Of 36 million variants and eQTLs, a certain number was selected that were located inside the 2118 chosen enhancers. What was this number? The sentence at line 375 seems convoluted. Does it mean that these "x" number of variants residing in 2118 enhancers were analyzed, based on the 'final ensemble' to predict their expression potential (prediction impact)? And then they were ranked based on this impact? What is the significance of this analysis for 500, 1000... variants? These variants were evaluated for their overlap with 1214 "true set" of regulatory variants from normal and cancer databases. The fraction of the variants that belong to the true set is the "precision", and that becomes the "model-based variant"? This section needs rewriting. Further, I do not see strong relevance for the location-based and random analysis. Besides, what meaningful information does this analysis for up to 5000 variants provide towards understanding the very fundamental question this manuscript aimed to understand: knowing the mechanistic difference between the +ve and -ve enhancers?

b. More than what is depicted in Fig 5c-e, it would have warranted greater interest and logic to show (1) that GEMSTAT predicts different expression or TF/CoReg binding of the variant enhancer sequences compared to a reference sequence, and (2) how the predicted differences correlate with existing variant (disease)-specific transcriptional datasets. For example, if there are 50 GTEx variants (normal) and 50 cancer (PancanQTL) variants, (1) how does their prediction for eRNA transcription and TF/TF-TF pair binding compare to their "reference" sequence; (2) how does this prediction hold true for these variants

in actual transcriptome/cistrome data from matched cancer samples/cell lines if publicly available; (3) how do they score as being +ve or a -ve enhancer; and how do they score on in silico knockdown analysis? Can we predict nullifying a cancer-linked belligerent variant enhancer by targeting a set of TFs? This straight-forward analysis would avoid the meandering statistical gymnastics this section and Fig 5 is plagued with.

c. Fig 6 is in line with the above logic with three specific examples, that is nice to see, However, since the study excluded the ER α -bound promoters and focused on Li et al's enhancers, how did the CHPT1 variant promoter ended up here? (Not a serious one; I am just curious).

14. Overall:

a. The initial goal of understanding the distinguishing features of the -ve and +ve enhancers got lost midway. Since eRNAs have therapeutic utility (PMID: 31594934), can a set of TFs be identified targeting which specifically a subset of +ve enhancers are blocked? Such a computational attempt would provide greater credence – even if no significant outcome is found, and that should be acceptable as part of the revised manuscript (a negative result is also a result; it advances science and, hence, the society).

b. Methods and definitions of terms are not adequately/elaborately described.

Response to Reviewer #1

Comment: In this manuscript, the authors report their computational analyses of estrogen receptor alpha (ERalpha)-bound enhancers. They chose ERalpha-bound enhancers using the ChIP-seq data of ERalpha and H3K27ac, and then divided them into positive and negative data based on their expression of enhancer RNAs. Combining with a set of known relevant TFBS information, they chose important sequence features using the random forest algorithm and they constructed simpler prediction models with fewer parameters. These models were further used for in-silico perturbation experiments and the prioritization of breast cancer-related sequence variants. Finally, the predicted effects of some variants were tested by the dual-luciferase reporter assay. Overall, I think that this is **an interesting computational study though newly-obtained biological insights may not be so significant** (in this sense, I feel that the latter part of their results could be shortened).

Here are some of the points I would like the authors to consider to improve the current manuscript:

1. Although they first attempted to build a thermodynamic model for discriminating eENA-producing and non-producing enhancers, the question on which are the main sequence differences were not explored well (nor the interpretation of their obtained thermodynamic model). Although it makes sense to show AUROC/AUPRC results, it may be biologically more important to show **how much the two classes of enhancers are different in terms of the presence/absence of several key motifs**.

Response: We thank the reviewer for their suggestion. We had previously provided (Figure 2C) a direct comparison of motif presence in the two classes of enhancers, for four key motifs – ER dimer, ER monomer, NR2F2, NR5A2. We have now added histogram plots illustrating the comparative distribution of affinity score for all considered motifs (Supplementary Figure S2, also reproduced below). As we see in these figures, in some cases there are statistically significant differences in distribution of motif-scores (predicted TF-DNA binding affinity) between the two classes of enhancers. However, none of these differences alone can fully discriminate between the two classes (overlap between histograms is substantial) and because of that we attempt to use a combinatorial approach (thermodynamic models) for this task.

Above: Reproduction of Supplementary Figure S2 from revised manuscript. See response above for interpretation.

Comment: 2. The reason why they took two steps in model building is not clear. It seems more straight-forward to use, say, LASSO directly. If they claim that their method is better, they should show the result of their comparison.

Response: As highlighted in a recent publication in NAR ¹, GEMSTAT has clear advantages over linear models, showcased within a well-understood system characterized by known motifs. Moreover, our extensive experience with GEMSTAT allows us to fine-tune the model to fit the task at hand. However, identification of relevant motifs from a larger compendium is not among GEMSTAT's features. Hence, we took the initial step to identify and narrow down the potentially important motifs. We would like to emphasize that our intention is not to introduce a new method (GEMSTAT has been used in over 10 publications) nor to assert any general superiority. Rather, we seek to present a mechanistic bioinformatic analysis of a specific data-rich biological system, using a tool designed for mechanistic interpretations.

Comment: 3. In Table 1, they claim that they could confirm the significance of their results through literature survey. However, they used a restricted set of TFs from the beginning and thus it does not make much sense.

Response: We thank the reviewer for pointing out the confusion. The literature support we claim in Table 1 relates to the general agreement between model's inferred *roles* (activator or repressor) for TFs (as depicted by the sign of change in predicted activity), and the experimental evidence regarding the TF's role reported in the cited papers. As the reviewer correctly mentions, we identified a set of TFs to include in our study through literature review. However, we did not pre-define the role of any TF or the nature of TF-TF interactions (i.e., cooperative or competitive) in this modeling exercise; these were automatically learned by the model.

Comment: 4. In my opinion, many of the figures are poorly presented: i) Fig.2a is not referred to in the Results and it does not explain the subsets they used in their study.

Response: We thank the reviewer for pointing this out. We have now modified the manuscript (Lines 173-178) to address this comment.

Comment: ii) In Fig.3f, the differences between the three groups are hard to recognize.

Response: We understand the difficulty in observing the difference in the three groups of models given the number of involved parameters. To address this issue, we have pointed out some of the differences among the three groups of models in the text in lines 281-290.

Comment: iii) In Fig.4d/e, which motifs are significant is hard to recognize.

Response: We thank the reviewer for pointing this out. We have modified figures 4d/e and their caption to show the pairs with statistically significant difference (Wilcoxon rank sum test adjusted p-value less than $1e-6$) more clearly.

Comment: iv) Figs.6a/d/g are hard to interpret.

Response: We thank the reviewer for pointing out the confusion. We modified the figure legend to address this point.

Comment: 5. In the results presented in Figs. 5c/d, they used a location-based model as a control, how about using a simple weight matrix-based scores within the enhancer regions, instead? In other words, how much better is their model than a simple WM-based model based on detected significant motifs?

Comment: The reviewer's suggestion does not offer a simple way forward for assessing a weight matrix (WM)-based baseline. To implement such a baseline, we have to score each variant for each WM motif and then figure out a way to compare impact scores among different motifs. Standard WM scores such as LLR are not comparable between WMs in practice, due to differences in their lengths and information contents. GEMSTAT can be thought of as a way to combine WMs for multiple TFs into one score for an enhancer.

Comment: 6. I wonder if their reporter assay experiments are supportive enough for their models. The results of changing the consensus sequence of known TFBSs sound somewhat trivial. How about, for example, changing the sequences of negative enhancers? In this sense, they should confirm if the negative enhancers are free from breast cancer-related variants.

Response: We apologize for the lack of clarity regarding the purpose of the reporter assay experiments. We did not change the consensus sequence of known TFBS. Rather, we changed the sequence of a *predicted* TFBS, which was predicted as being important to the enhancer using our model. We did not seek to show that the negative enhancers are free from BrCa-related variants; the reporter assays were designed to show that the model's interpretation of an existing variant's *mechanism of impact* (TFBS whose function is impacted) is supported by experimental perturbation.

Response to Reviewer #2

Comment: Overall impression of the work: This is an interesting work. eRNAs have predictive gene-regulatory power, and there are efforts to target eRNA expression in breast cancer therapeutics. So, knowing what sequence features of enhancers dictate eRNA expression and how this is dysregulated in disease-specific variants will have immense value. However, even as they seem to have started the work with this goal, the authors seem to have gone astray a bit (details below). This manuscript would be apt for the journal; but there are several concerns in the manuscript that must be addressed before the manuscript can be considered for acceptance. The authors are a team of accomplished scientists and addressing these concerns should be easy.

Response: We thank the reviewer for their favorable opinion of our manuscript and for their careful reading and constructive criticism and suggestions. We provide our point-by-point response below.

Comment: 1. Page 2: in the “conclusions” section, the authors simply state ‘gained mechanistic insights... program’. Please specify the concluding points/outcomes (mechanistic insights gained) of the work here.

Response: We thank the reviewer for their suggestion. Here we note some of the mechanistic insights gained by thermodynamics-based modeling exercise, which we have also added to the conclusion section in this revision:

“Based on in-silico knockdown analysis, we concluded that the models learned an activating role for ER α , FOXA1, GATA3, NR5A2, PAX2, PBX1, and RAR α . This activating role combines the binding strength and activation potential for each given TF. In case of ER α most of the models learned a strong activation parameter paired with a weak binding parameter. We also noted that the models learned strong cooperativity parameters between ER α and several TFs including FOXA1, GATA3, and RAR α , all of which are leaned as strong binders. The high activation parameter, and weak binding parameter learned for ER α , together with its strong cooperativity suggest a potential mechanism in which ER α possesses strong transcriptional activation potential but a weak DNA binding capability, and thus utilizes co-factors such as FOXA1 and GATA3 for DNA binding. Additionally, most models assign a repressive role to Glucocorticoid Receptor (GR). Similarly, almost all models learn a strong cooperative interaction between ER α and RAR α . The interplay between ER α , YBX1, and RUNX1 can be an interesting future direction to explore.”

Comment: 2. Page 7: This work is based on the ChIP-seq/GRO-seq data sets of Li et al. 2013. However, the numbers do not seem to match. Li et al identified 7174 potential enhancers enriched for ER α , H3K4me1 and H3K27ac upon E2 treatment; of these, 1045 occur within 200kb of a gene TSS upregulated by E2 (of these 1033 bind ER α only at the enhancers; see Fig S1). Thus, these 1145 were considered as E2-responsive enhancers.

a. So, where does the number 3734 as ‘potentially active enhancers’ come from? Do the authors use different thresholds to analyze sequence reads? This needs clarification.

Response: We apologize for the lack of clarity on this, and have now clarified the procedure in Methods (lines 719-724):

“We considered the full set of 3734 ER α binding sites with H3K27Ac marks in estrogen-treated MCF7 cell line (Li et al. ²) as the “universe set” of genomic segments to analyze. To identify 3734 potentially active enhancers, we used H3K27Ac and ER ChIP-seq processed data (in bigwig format) published by the authors (with a threshold score of 5 on peak scores and merged peaks within 100 bp of each other). We took the intersection between the peaks found in the two datasets to get to the set of 3734 potential enhancers.”

Comment: b. Of the above 3734, the authors designated 449 as +ve (for E2-induced eRNA expression) and 1669 as -ve enhancers (no E2-induced eRNA), and the rest are ignored. This is a curious case. How do the authors quantify eRNA? Since (1) eRNAs are thought to mark “active” enhancers (that are implicated in activation of a target gene undergoing transcription), and (2) the magnitude and temporal kinetics of eRNA production correlate with the production of the target mRNAs (thus, the magnitude of eRNA expression can be reflective of an enhancer’s activity, as well as of the target gene’s transcription, and the authors seem to accept this assumption), the 3734 ‘potentially active enhancers’ are expected to (1) activate target gene(s) and (2) produce eRNAs. How do these enhancers contain 1669 that do not transcribe? If they do not transcribe eRNAs, are they still activating target genes? Or, are the authors hinting at a set of active enhancers activating target genes yet not transcribing themselves? That would be a paradigm-shifting notion. And, if so, do these enhancers engage in contact with the target ‘transcribing’ promoters while themselves not transcribing? Further, what are the “ignored” enhancers? Since all enhancers considered here are ER α -bound, they all can come in only two possible classes: either transcribing eRNAs or not transcribing eRNAs. What is this third class that was ignored? These questions are relevant and need answers since these for the very premise that the entire manuscript rests upon.

Response: We thank the author for pointing out the confusion. In this manuscript we aim at modeling transcription of enhancers (eRNA), as a marker of enhancer activity. We have not tried to model mRNA expression (of genes presumably regulated by the enhancer) and have not made any claims in that regard. Among the 3734 potentially active enhancers (defined solely based on H3K27Ac mark and ER binding), a subset produces eRNA, and a subset does not. We expect that the subset that does not produce eRNA to be inactive and to not influence gene expression, though this remains speculation and our modeling does not depend on the validity of that hypothesis. We have used processed datasets from Nascent Transcription Repository to identify e-RNA transcribing regions.

We have added a clarifying section to the manuscript lines 172-178 to explain why we have excluded some potential enhancers from this analysis:

“Focusing specifically on enhancers affected by estrogen, we excluded 40 out of 489 eRNA-overlapping examples shown in Figure 2a because those regions overlapped with GRO-Seq peaks both before and after estrogen treatment. Additionally, keeping the noisy nature of GRO-Seq data in mind, to avoid false negatives in our classification analysis we excluded 1576 out of 3245 non-eRNA-overlapping examples shown in Figure 2a because of overlap with GRO-Seq peaks in one of the two replicate experiments.”

Comment: c. For the benefit of the readers and reviewers alike, I ask that the list of the coordinates of the +ve, -ve and “ignored” enhancers (and preferably their potential target genes) is provided as a supplemental file, along with their relative E2-induced sequence reads (or fold-change) of the eRNAs and corresponding target mRNAs.

Response: We have now provided the coordinates of positive and negative class enhancers in Additional Files 6, and 7.

Comment: 3. Page 7/8: Figures 1 and 2a are not cited. Please make sure that all figures are cited in the main text in the order they are numbered.

Response: We have addressed this point in the revised manuscript (Lines 136, 163).

Comment: 4. Page 8, line 177: What is a ‘numeric feature’? How were 594 of them derived? This should be explained in the methods or in a supplemental text. At the end of this paragraph, the authors select 16 of these numeric features, which are listed as TFs in Table S1. So, is a TF considered a numeric feature? Why? What were the criteria to select “16 most important features” out of these? These need to be explained somewhere.

Response: We thank the reviewer for pointing out the confusion regarding the 594 features describing each enhancer. We have made a few clarifications to the text in the method sub-section titled “Feature set construction” to address this confusion. The full set of 594 features consisted of 33 features representing individual motif scores (as measure by the likelihood ratio score for the motif, thus a “numeric” feature), together with 561 features representing the pairwise adjacency of motifs (number of occurrences of each pair of TF motifs on the enhancer within a 50 bp distance of each other). These 594 features were used to train the random forest model. We then used the importance score from the random forest model to rank the motifs in terms of their importance to model predictions, and used the top 16 important motifs to train the GEMSTAT models.

Comment: 5. Page 8, line 178: How was the “adjacency score” calculated? No explanation anywhere. What does a -ve value mean (Fig 2b)?

Response: In lines 714-718 of the revised manuscript, we explain how the adjacency score was computed. We have included the relevant text below for the reviewer’s convenience.

“In order to construct features capturing pairwise TF interactions, we set an LR score threshold for each TF corresponding to a p-value of $1e-4$, and marked putative sites stronger than this threshold. Next, for all pairwise combinations (total of 561 pairs) of the 33 used motifs, including homotypic-pairs, we counted the number of adjacent sites (within 50 bp of each other) in an enhancer; this count was used as the pairwise adjacency feature of the TFs.”

Comment: 6. Page 8, line 183: There is insufficient methods on how AUROC and AUPRC values were derived, and no explanation on how to interpret their numeric values. The authors should consider the fact that numerous researchers follow classical gene regulation and hormone regulation who may not necessarily have exposures to the authors’ line of computational approaches.

Response: These are two complementary metrics to assess sensitivity and specificity of predictive models across the full spectrum of prediction stringency. Intuitively, AUROC (Area Under Receiver Operating Characteristics) represents the probability that the classifier ranks any randomly chosen positive example higher than any randomly chosen negative example. We also used AUPRC (Area Under Precision Recall Curve) as this metric is better suited to evaluate performance of classifiers on datasets where the number of examples in the positive class is

smaller than the number of examples in the negative class (i.e., imbalanced dataset)³. Baseline AUROC for any binary classification task is equal to 0.5. The baseline AUPRC depends on the dataset and is equal to the number of positive examples divided by the total number of examples. As mentioned in line 195 of the revised manuscript random baseline AUPRC for our dataset is equal to 0.21. We have provided these random baseline values of the AUROC and AUPRC in the text and cited a relevant paper for more details on these metrics, which are commonly used in the bioinformatics field.

Comment: 7. Page 9, line 203: Reference 25 is the same as reference 21. Please remove the duplicate.

Response: Thanks for catching this error. We have now addressed this in the manuscript.

Comment: 8. Page 9, line 204: More description needed on GEMSTAT as adopted here. The current GEMSTAT analysis is based on the initial analytic pipeline in He et al, 2010, where mathematical modeling included thermodynamic properties of TFBSs, cellular abundance of TFs, known or predicted TF-TF and TF-basal transcription machinery interactions in Drosophila. The present scope of GEMSTAT seems much expansive compared to the Drosophila work, where only a handful TFs were considered. Plus, the current work appears heavily compromised as it does not consider TF availability and Pol II occupancy at the enhancers. Quantitative TF availability can be found elsewhere (for example, Nusinow et al. Cell, 180: 387; 2020), while the authors can conduct their own Pol II ChIP-seq if not available with Li et al, 2013. Another complexity could “potentially” compromise the predictive power: the TF occupancies at the promoters – both core promoter elements and proximal promoter TFBSs, plus Pol II (it would be naïve to think that the enhancer alone determines the transcriptional output of a gene, with no contribution from the promoter). While I understand that these expansions in the analytic framework may not be possible at the moment, it nevertheless makes this manuscript look undercooked.

Response: We thank the reviewer for pointing out this potential pitfall of our study. As pointed out in lines 689 to 696 of the revised manuscript, these points form important future directions of the current study:

“However, due to the lack of comprehensive data on TF concentration measurements in the E2 treated MCF7 cell lines, our models currently examine only sequence differences among enhancers and are limited to making predictions in conditions similar to the training condition. Despite these limitations, GEMSTAT offers a way to interpret enhancer sequences while accommodating sequence properties relevant to function, e.g., numbers and strengths of binding sites, overlaps between sites, cooperativity, and antagonism between TFs. Future extensions of our work will address this modeling limitation, e.g., by inference of effective TF concentrations from available ChIP-seq data.”

GEMSTAT was developed to utilize TF concentrations if available, but these concentrations matter to the model only if there are multiple cell states/types/conditions being modeled; in such a case the relative levels are important. For modeling TF influences in a single cellular context, TF concentrations are not required, and even if such data were available, they will not change the model training, as there is a free parameter per TF to scale the relative concentration to absolute levels.

Core promoters are not considered in the GEMSTAT framework and to our knowledge no biophysics-based model exists (GEMSTAT or otherwise) that mimics the enhancer-promoter

interactions in modeling gene expression. Moreover, since we are modeling eRNA expression (versus not), it is not clear what promoter occupancy may mean.

Comment: 9. Page 9, line 217: How are the “17 TFs and 9 TF-TF interactions” selected and why? Are these related to the 16 numeric features of Table S1?

Response: We thank the reviewer for pointing out the lack of clarity on the choice of TFs and TF-TF interactions used for GEMSTAT modeling. We have clarified this selection on lines 229-237, excerpted below for the reviewer’s convenience.

“We chose 14 most highly ranked TFs based on Random Forest importance score (Table S1 in Additional File 1), together with three other TFs known to be involved in the ER transcriptional program (i.e., GATA3, GR, and RAR α) to form the set of TFs used for GEMSTAT modeling. Additionally, based on the literature and the trained random forest model, we allowed nine TF-TF interaction terms to be included in the GEMSTAT models. These interaction terms consist of interactions of ER α with its potential co-factors FOXA1, GATA3, PBX1, PGR, RAR α , AP2- γ , together with a self-interaction pair for JUN-1 allowing for its self-cooperativity, as well as two most highly ranked TF-TF pairs in terms of importance score by the random forest models (i.e., ER α -YBX1, and RUNX1-YBX1).”

Comment: 10. Page 10; also Fig 3: What criteria were used to identify and differentiate the “binding” and “activation” parameters, and what does a -ve value mean here (Fig 3f)? This needs to be explained. Classically speaking, the binding and activation functions of a TF are tricky to separate. A TF may bind the target site strongly but may not strongly activate the gene, but a strong activator is “likely” to show stronger binding. This is where this analysis does not inspire confidence, because of conceptual inconsistency. Here are some examples:

Response: We “log 10 transformed” the parameter values to make the visualization clearer. Negative values show parameters with value less than one. We have added text to figure 3 caption to make this clearer.

Separation of activation and binding parameter is tricky but possible. This is done in each of the GEMSTAT papers. Recently, we published a careful examination of this aspect ¹. That said, there is indeed a level of “parameter coupling” possible, wherein the separate values of activation and binding are not properly resolved for some TFs. For this reason, the ensemble modeling is an important part of examining model parameters, as first introduced in Samee et al.⁴.

Comment: a. Despite the entire dataset based on ER α -bound enhancers for ER α -dependent transcriptional activation, Fig 3f shows no enrichment in ER α binding. How is it possible that analyses critically and centrally based on ER α -enriched datasets do not exhibit ER α enrichment?

Response: Figure 3f does not make any statement about enrichment of ER α binding in enhancers. (Enrichment, by definition, must pertain to one class of entities in comparison to another class; all entities in Figure 3F, shown as rows, are enhancers. No other class is shown.) This figure shows the parameter values of trained GEMSTAT models. Each row represents a top-performing model and each column represents a parameter. Binding parameter learned for ER α is close to zero for almost all the trained models. Since values are log10 transformed, this translates to a weak binding parameter of ~ -1 learned by all models for ER α . A mechanistic explanation is that ER α requires co-factors such as FOXA1 and GATA3 (which both have strong binding parameters learned by most models) for binding to DNA. Note that all models have learned a strong cooperativity parameter between ER α and both FOXA1 and GATA3.

Comment: b. Despite being a strong activator (Fig 4c), PBX1 scores very poorly for activation in Fig 3f. Why this discrepancy in data?

Response: PBX1 is indeed a strong activator according to Fig 4E. This is consistent with a strong binding parameter learnt for it (Figure 3F) and an activation parameter value greater than 1. Note that the parameter values are log 10 transformed for visualization in figure 3f. Thus a positive value for activation in figure 3f corresponds to an activation parameter of greater than one, which indicates activating role. However, for an activator to have an effect it must be itself bound to the enhancer. Binding strength is determined by the binding parameter of the TF and its cooperators TFs. The binding parameter includes the concentration scaling, i.e., the optimal site of a “strong binder” can have fractional occupancy close to 1 while optimal site of a “weak binder” can have much smaller fractional occupancy. A strong binder can score poorly for activation and yet exhibit strong activating effect in Fig 4E because a “knockdown” assesses the net effect of the TF. In summary, there is no discrepancy in the stated observations regarding PBX1.

Comment: c. YBX1, a strong repressor (Fig 4c; Table S2) shows in strong binding in group 1 and 3 enhancers with less repression, yet binds feebly to group 2 enhancers with stronger repression (Fig 3f). This appears to defy logic.

Response: Note that the groups noted in figure 3f correspond to clusters of GEMSTAT models with similar parameter values, and do not correspond to groups of enhancers. It is common in ensemble modeling to find sub-groups of models with different parameter settings; this helps interpret the remaining uncertainties in the modeling. For instance, we see here that all three sub-groups of models are consistent with a strong repressive role for YBX1, but two sub-groups (1 and 3) assign this strong role via a strong binding parameter while the third sub-group (2) assigns the strong role via an activation parameter $\ll 1$ (interpreted as strong repressive potential).

Comment: d. AP2- γ does not bind the enhancers (its binding score is -ve; but is there an alternative interpretation?), yet has moderate activation function – though is listed as a strong activator (Table 1).

Response: Note that the parameter values are log 10 transformed for visualization in figure 3f. AP2- γ has a strong activation parameter, and a weak binding parameter. However models have learned a fairly strong cooperativity parameter between AP2- γ and ER α , which can explain its activation properties. The net binding strength of a TF can be shaped by cooperative binding with other TFs.

Comment: e. FOXA1 “activates a large fraction, nearly 50%, of examined enhancers” (page 12, line 291; and Fig 4), and yet scores around zero for activation in Fig 3f.

Response: As can be seen in Figure 3f, most models learn a strong binding parameter for FOXA1, as well as a strong cooperativity parameter between FOXA1 and ER α . This is in fact consistent with the pioneer factor properties of FOXA1⁵. It binds strongly to enhancers and recruits ER α which is an strong activator through cooperative interaction. The recruitment of ER α and its strong learnt activation parameter explain the net activatory effect of FOXA1 as shown in figure 4e.

Comment: 11. Page 11: The validity, relevance and significance of the in silico perturbation experiments are unclear. Here is why. The RF and GEMSTAT analytic pipelines were trained for enrichments of certain TFs. So, when some of those TFs would be omitted in in silico perturbation,

it is natural that the readouts will be adversely affected; there is no novelty here. For example, a dataset trained on an activator will predict loss of activation when the activator is omitted. Instead, the authors need to conduct real MCF7-based knockdown experiments and show that:

- A TF upon knockdown leads to loss of transcription from an enhancer it is predicted to activate
- In the same experiment, those enhancers are unaffected where the TF is not predicted to activate.

I believe the authors' GEMSTAT and Regulon analyses can likely predict even secondary regulatory effects.

Response: The in silico perturbation experiments are carried out to characterize the regulatory effects of each TF – whether it is an activator or repressor, how many eRNAs are influenced by it and to what extent, etc. The reviewer's phrase "For example, a dataset trained on an activator will predict loss of activation when the activator is omitted" suggests a possible misunderstanding – the models were not trained with prior knowledge of activation or repression roles, nor their strengths of regulation; thus, doing the in silico perturbations provides a way to assess the direction and extent of those roles. While the TFs used in the model were indeed selected based on prior evidence of their relevance to the ER system, that prior knowledge is not necessarily detailed enough to provide the direct regulatory role (activator/repressor) or regulons (target genes) of the TF. Similar in-silico knockdown analysis is performed by Bhogale et al. ⁶

Comment: 12. Page 15, lane 350 (also page 27, line 639): I strongly suggest omission of the term "universe". Scientific communications should not encourage scientifically incorrect symbolism. "Universe" essentially means "all that there is, was, will be – ever" and goes unfathomably beyond the "observable universe", which has a radius of over 46 billion light years. It is all-inclusive – known or unknown, with absolutely no exception. However, the usage "universe" of enhancers is so specifically exclusive minuscule an entity here that it does not even include 'all potential enhancers in human genome – active or otherwise', let alone all enhancers ever evolved in any organism. Such a narrow usage will be a disrespect to our universe.

Response: We used the term "universe" in the common statistical sense that it is used to describe a Hypergeometric test, or in the mathematical sense in set theory. The universe refers to the background set of which two subsets are considered and whose mutual overlap is assessed for significance under the Hypergeometric distribution. See [https://en.wikipedia.org/wiki/Universe_\(mathematics\)](https://en.wikipedia.org/wiki/Universe_(mathematics)) for example. We have now made this point clearer by using the phrase "universe set" rather than just "universe", though the latter is also in common usage.

Comment: 13. Page 16: Mechanism-aware variant impact prediction:

a. It is not readily clear what is a "model-based variant" (red boxes in Fig 5c-e). Of 36 million variants and eQTLs, a certain number was selected that were located inside the 2118 chosen enhancers. What was this number? The sentence at line 375 seems convoluted. Does it mean that these "x" number of variants residing in 2118 enhancers were analyzed, based on the 'final ensemble' to predict their expression potential (prediction impact)? And then they were ranked based on this impact? What is the significance of this analysis for 500, 1000... variants? These variants were evaluated for their overlap with 1214 "true set" of regulatory variants from normal and cancer databases. The fraction of the variants that belong to the true set is the "precision", and that becomes the "model-based variant"? This section needs rewriting. Further, I do not see strong relevance for the location-based and random analysis. Besides, what meaningful information does this analysis for up to 5000 variants provide towards understanding the very fundamental question this manuscript aimed to understand: knowing the mechanistic difference between the +ve and -ve enhancers?

Response: Model-based variants are the ones predicted by the models to significantly affect the activity of the enhancer they reside in. Out of about 36 million total relevant variants, 97529 variants were located inside the 2118 enhancers considered in this study. Coordinates of those variants is now provided in Additional File 8.

We investigated the predicted effect for each of the 97529 variants on the activity of their host enhancer. To do this we predicted the expression driven by each enhancer incorporating each of the variants. We quantified the predicted effect of the variant by comparing the predicted activity of the mutated enhancer (incorporating the variant) to its WT version using the final ensemble of GEMSTAT models. We ranked the variants based on their predicted effect and considered K (500, 1000, ..., 5000) top-ranking to evaluate accuracy. Note that the accuracy of such models is usually evaluated at a range of thresholds to avoid one arbitrary threshold setting.

An example may help to clarify the precision calculations. For K equal to 500, we evaluate the precision of the variant set as the number of variants in the set that overlap with 1214 known functionally relevant variants divided by the size of the set (i.e., 500).

We have now reworded the relevant passage of text to improve the clarity of presentation, and we excerpt that entire passage below for the reviewer's convenience.

"We prioritized variants from a large collection of over 36 million common variants and eQTLs. Out of about 36 million total relevant variants, 97529 variants were located inside the 2118 enhancers considered in this study (described in Additional File 8). We investigated the predicted effect for each of the 97529 variants on the activity of their host enhancer. To do this, using each of the 244 GEMSTAT models in the final ensemble, we predicted the expression driven by mutant enhancers incorporating each of the variants. Using each GEMSTAT model, we quantified the predicted effect of the variant by comparing the predicted activity of the mutant enhancer (incorporating the variant) to its WT version. We finally short-listed ("prioritized") the top (say K = 500, 1000, ... , or 5000) variants ranked by predicted impact (i.e., model-based variants). These variants were then evaluated by their overlap with a "true set" of 1214 regulatory variants comprising breast-related eQTLs from GTEx and breast-cancer related eQTLs from the PanCanQTL database, by noting down the fraction of prioritized variants that belong to the true set. We call this fraction the "precision" of the prioritized set. An example may help to clarify the precision calculations. For K equal to 500, we evaluate the precision of the variant set as the number of variants in the set that overlap with 1214 known functionally relevant variants divided by the size of the set (i.e., 500). Repeating this procedure with each of the 244 GEMSTAT models in the final ensemble, we obtained as many estimates of precision of this model-based variant prioritization approach, shown in each red box of Figure 5c (for different values of K). To assess the significance of model-based variant scoring, we then performed the same evaluations with a "location-based" approach where K variants located within the enhancers were selected at random rather than based on predicted impact, thus utilizing functional information about enhancer locations but not the extra information provided by models of those enhancers. Again, by repeating the random sampling 244 times we obtained as many estimates of precision of this simpler strategy (shown as blue boxes in Figure 5b). Finally, as a baseline approach, we selected K variants at random from the entire collection, without regard to location within enhancers or model-based predictions, and obtained a distribution over their precision score (by repeating the sampling 244 times). These are shown as green boxes in Figure 5c."

Comment: b. More than what is depicted in Fig 5c-e, it would have warranted greater interest and logic to show (1) that GEMSTAT predicts different expression or TF/CoReg binding of the variant enhancer sequences compared to a reference sequence, and (2) how the predicted

differences correlate with existing variant (disease)-specific transcriptional datasets. For example, if there are 50 GTEx variants (normal) and 50 cancer (PancanQTL) variants, (1) how does their prediction for eRNA transcription and TF/TF-TF pair binding compare to their “reference” sequence; (2) how does this prediction hold true for these variants in actual transcriptome/cistrome data from matched cancer samples/cell lines if publicly available; (3) how do they score as being +ve or a -ve enhancer; and how do they score on in silico knockdown analysis? Can we predict nullifying a cancer-linked belligerent variant enhancer by targeting a set of TFs? This straight-forward analysis would avoid the meandering statistical gymnastics this section and Fig 5 is plagued with.

Response: We thank the reviewer for this suggestion. We have pursued what the reviewer is suggesting for three selected variants and even followed it up in experimental reporter assays as depicted in next section of the manuscript.

Comment: c. Fig 6 is in line with the above logic with three specific examples, that is nice to see, However, since the study excluded the Era-bound promoters and focused on Li et al’s enhancers, how did the CHPT1 variant promoter ended up here? (Not a serious one; I am just curious).

Response: In this study we did not exclude promoters. We chose the regions of interest based on the overlap of H3K27Ac mark and ER α ChIP-Seq peaks. H3K27Ac is a known marker for enhancers but may be observed on promoters as well. The CHPT1 promoter analyzed in this study overlapped with H3K27Ac and ER α ChIP-Seq peaks.

Comment: 14. Overall: a. The initial goal of understanding the distinguishing features of the -ve and +ve enhancers got lost midway. Since eRNAs have therapeutic utility (PMID: 31594934), can a set of TFs be identified targeting which specifically a subset of +ve enhancers are blocked? Such a computational attempt would provide greater credence – even if no significant outcome is found, and that should be acceptable as part of the revised manuscript (a negative result is also a result; it advances science and, hence, the society). b. Methods and definitions of terms are not adequately/elaborately described.

Response: We hope our point-by-point responses above and the associated changes in the manuscript have helped improve the manuscript in the reviewer’s opinion. We are extremely grateful for your thoughtful, constructive and detailed evaluation of our work.

References

1. Dibaeinia, P. & Sinha, S. Deciphering enhancer sequence using thermodynamics-based models and convolutional neural networks. *Nucleic Acids Res.* **49**, 10309–10327 (2021).
2. Li, W. *et al.* Functional roles of enhancer RNAs for oestrogen-dependent transcriptional activation. *Nature* **498**, 516–520 (2013).
3. Saito, T. & Rehmsmeier, M. The precision-recall plot is more informative than the ROC plot when evaluating binary classifiers on imbalanced datasets. *PLoS One* **10**, e0118432 (2015).
4. Samee, M. A. H. *et al.* A Systematic Ensemble Approach to Thermodynamic Modeling of Gene Expression from Sequence Data. *Cell Syst.* **1**, 396–407 (2015).
5. Seachrist, D. D., Anstine, L. J. & Keri, R. A. FOXA1: A Pioneer of Nuclear Receptor Action in Breast Cancer. *Cancers (Basel)*. **13**, (2021).
6. Bhogale, S. & Sinha, S. Thermodynamics-based modeling reveals regulatory effects of indirect transcription factor-DNA binding. *iScience* **25**, 104152 (2022).

Reviewers' comments:

Reviewer #1 (Remarks to the Author):

Although the authors addressed many of my points, I do not think that some of their revisions are enough:

Follow up on Comment 1.

Response to Comment 1: We thank the reviewer for their suggestion. We had previously provided (Figure 2C) a direct comparison of motif presence in the two classes of enhancers, for four key motifs – ER dimer, ER monomer, NR2F2, NR5A2. We have now added histogram plots illustrating the comparative distribution of affinity score for all considered motifs (Supplementary Figure S2, also reproduced below). As we see in these figures, in some cases there are statistically significant differences in distribution of motif-scores (predicted TF-DNA binding affinity) between the two classes of enhancers. However, none of these differences alone can fully discriminate between the two classes (overlap between histograms is substantial) and because of that we attempt to use a combinatorial approach (thermodynamic models) for this task.

I am sorry, but the answer provided by the authors is not enough. In my understanding, one of the main advantages of using GEMSTAT, instead of other more complex models such as deep learning ones, would be its interpretability. Moreover, GEMSTAT is introduced as a sequence-to-expression model and combinatorial approach. Then, the interpretation in terms of the input features (motifs) on how the model is able to classify expressing and non-expressing enhancers is rather necessary. If single motifs are not sufficient, cooperative motifs highlighted by the model as important in the classification task should be explored.

Follow up on Comment 4. The Figures are still poorly presented. In the current version of Figures, carefully reading the legends and manuscript is entirely necessary to understand them. Instead, the authors should prepare self-explanatory Figures. Take the next examples (many more can be mentioned):

Response to Comment 4ii: We understand the difficulty in observing the difference in the three groups of models given the number of involved parameters. To address this issue, we have pointed out some of the differences among the three groups of models in the text in lines 281-290.

More than adding extra explanations to the text, which is already considerably long, the authors should improve their Figures, so a not long explanation is necessary. For instance, in Figure 3f, except for YBX1 and its interactions with other motifs, most of the other motifs don't show a strong difference among groups. Then, only the significant information to differentiate the groups should remain as a main Figure, and the extra information should be included as a supplementary Figure.

Response to Comment 4iv: We thank the reviewer for pointing out the confusion. We modified the figure legend to address this point.

Again, instead of extending the description, a self-explanatory Figure is preferred. For example, instead of explaining in the Figure legend that we are looking at the 1Kbp region, why not add this information to the Figure? which would be relatively easy to change. Also, in the current version, it's not clear where the region starts/ends; a track delimiting the region is necessary. Also, more information on the tracks should be added; for example, "somatic variants" should be used instead of "variants".

Extra comment. After reading the current version of the manuscript, a new question came up. In Figure 4d-e, the authors compared the effect of in-silico knocking down single and cooperative motifs.

The authors presented the percentage of affected enhancers, positive and negative classes, and the mean percentile change in expression. The change in expression of negative classes is confusing for me as, in principle, negative samples lack expression levels; eRNA expression. Are the authors comparing eRNA expression or other epigenetic signals? This should be clarified in the Figure and the manuscript.

Reviewer #2 (Remarks to the Author):

I have reviewed the revised version of the manuscript COMMSBIO-22-1690A by Tabe-Bordbar et al., titled "Mechanistic analysis of enhancer sequences in the Estrogen Receptor transcriptional program". I had raised several points in the initial review, and I am satisfied that the authors have addressed most of those issues. I think the manuscript can be accepted now. I look forward to seeing this manuscript in print. I think that the work presented here can be expanded to include many other parameters in future that can provide deeper understanding of sequence features encoded in enhancers work towards stimulating target transcription.

Thank you,
Sincerely
Reviewer 2

Comment: In my understanding, one of the main advantages of using GEMSTAT, instead of other more complex models such as deep learning ones, would be its interpretability. Moreover, GEMSTAT is introduced as a sequence-to-expression model and combinatorial approach. Then, the interpretation in terms of the input features (motifs) on how the model is able to classify expressing and non-expressing enhancers is rather necessary. If single motifs are not sufficient, cooperative motifs highlighted by the model as important in the classification task should be explored.

Response: The reviewer requests a “interpretation in terms of the input features (motifs) on how the model is able to classify expressing and non-expressing enhancers”. To us, the best way to probe “how the model is able to classify” is to show the effect of removing each motif’s presence in an enhancer and noting its effect on the model’s prediction for that enhancer. We had summarized such information in Figure 4e, and have now added a heatmap (Supplementary Figure S4, reproduced below for reviewer’s convenience) where this information is presented for all enhancers and all motifs. For each enhancer, this heatmap thus tells us which motifs (putative TF binding sites) played a strong or weak activating or repressive role. To us, this is the simplified interpretation of the trained GEMSTAT model. (The same information is also presented as a supplementary table in additional File 9, for easy readability.)

Please note that our notion of “interpretability” is that the GEMSTAT model’s prediction is made via a well-established (interpretable) function grounded in biophysical principles. We used GEMSTAT because the model itself is interpretable in the language of molecular interactions (free energies, TF concentrations, etc.) and its parameters have simple biophysical interpretations. Our use of the word “interpretability” is somewhat different from that of the modern Machine Learning literature, where it is often used to refer to making a complex black-box model understandable to the layperson in terms of its feature usage.

We also note that the newly added heatmap/table only approximately explains which motifs (TFs) contributed to an enhancer’s predicted expression, not how those motifs act together as part of a whole – the latter is explained via GEMSTAT’s thermodynamics-based model. Also, Figure S4 is an attempt to provide a high-level view that summarizes the enhancer-level effects of motifs across all enhancers; and while it shows some salient aspects of the overall model (e.g., motifs that contributed significantly to several enhancers), such an aggregate view is bound to lose resolution in describing the model’s functioning on individual enhancers.

Figure S4. TF removal effect on individual enhancers. The heatmap illustrates the average percent change in predicted expression of the examined enhancers (columns), after removal of each considered TF (rows). Red and green column side color bars show negative and positive class enhancers, respectively.

Comment: More than adding extra explanations to the text, which is already considerably long, the authors should improve their Figures, so a not long explanation is necessary. For instance, in Figure 3f, except for YBX1 and its interactions with other motifs, most of the other motifs don't show a strong difference among groups. Then, only the significant information to differentiate the groups should remain as a main Figure, and the extra information should be included as a supplementary Figure.

Response: We thank the reviewer for pointing out the difficulty in interpreting figure 3f. We should note that the purpose of this figure is to provide an overall visualization of the ensemble of thermodynamic models. More specifically, it aims to show how the activation, binding, and cooperativity parameters can be used to directly interpret the model findings. Additionally, we aimed to show how the different models agree on some parameters while disagree on others. However, we understand that visualizing all parameters at the same time hinders interpretability and therefore removed 10 of the parameters from the main figure and made some adjustment to the labels to improve this figure. The modified figure is reproduced here for the reviewer's convenience. We have provided the full parameter visualization as Figure S3.

Comment: (regarding Figure 6) Again, instead of extending the description, a self-explanatory Figure is preferred. For example, instead of explaining in the Figure legend that we are looking at the 1Kbp region, why not add this information to the Figure? which would be relatively easy to change. Also, in the current version, it's not clear where the region starts/ends; a track delimiting the region is necessary. Also, more information on the tracks should be added; for example, "somatic variants" should be used instead of "variants".

Response: We appreciate the reviewer's constructive comments and have implemented their comments in figures 6 a,d, g. A panel delimiting the region is added and legend has been modified accordingly. We should note that the considered variants in this figure consist of somatic, common germ-line and eQTL SNPs, hence we changed the name from Variants to SNPs. The modified figure is reproduced here.

Comment: Extra comment. After reading the current version of the manuscript, a new question came up. In Figure 4d-e, the authors compared the effect of in-silico knocking down single and cooperative motifs. The authors presented the percentage of affected enhancers, positive and negative classes, and the mean percentile change in expression. The change in expression of negative classes is confusing for me as, in principle, negative samples lack expression levels; eRNA expression. Are the authors comparing eRNA expression or other epigenetic signals? This should be clarified in the Figure and the manuscript.

Response: We thank the reviewer for pointing out the confusion. In this figure we present the results of *in silico* knock down analysis. In this exercise, we null parameters from the models one at a time and note the change in predicted activity of enhancers. However, we understand that the figure does not clearly indicate the presentation of predicted values as opposed to experimentally measure values. To make this clear, we have modified the Y axis label in figure 4e from “Mean percentile change in expression” to “Mean percentile change in predicted expression”. We also changed the Y-axis label in figure 4d from “% enhancers affected by more than 5 percentile” to “% enhancers with higher than 5 percentile change in predicted expression”. Additionally, we added the following paragraph to the manuscript (Lines 275-280).

“Note that the models make a numeric prediction for the activity of both positive and negative class enhancers. In case of a perfectly accurate model, the predicted values for all negative class enhancers are less than that of all positive class enhancers. The predicted activity can be compared before and after removal of a certain TF (i.e., in-silico knock-down) to quantify the TF’s predicted effect on enhancer activity as learned by the models.”

REVIEWERS' COMMENTS:

Reviewer #1 (Remarks to the Author):

The authors mostly addressed my previous comments; I thank them for their work. Regarding my previous comment, I understand the limitations of GEMSTAT on the interpretability of motifs. Thus, I would like to suggest the authors to describe this limitation further in their discussion. I understand that they already mentioned this in lines 649 to 652. However, a clearer and more direct statement would be preferred, stating the limitation of GEMSTAT to identify de-novo motifs or motifs of TF not included in the training dataset, although I do not mean this proposal is mandatory.